# CAN KNOWLEDGE EDITING REALLY CORRECT HALLUCINATIONS?

**Baixiang Huang**[*1]**, Canyu Chen**[*2]**, Xiongxiao Xu**[2]**, Ali Payani**[3]**, Kai Shu**[†1]
[1]Emory University, [2]Illinois Institute of Technology, [3]Cisco Research
{baixiang.huang,kai.shu}@emory.edu,{cchen151,xxu85}@hawk.iit.edu,apayani@cisco.com

Project website: https://llm-editing.github.io

## ABSTRACT

Large Language Models (LLMs) suffer from hallucinations, referring to the non-factual information in generated content, despite their superior capacities across tasks. Meanwhile, knowledge editing has been developed as a new popular paradigm to correct erroneous factual knowledge encoded in LLMs with the advantage of avoiding retraining from scratch. However, a common issue of existing evaluation datasets for knowledge editing is that **they do not ensure that LLMs actually generate hallucinated answers to the evaluation questions before editing**. When LLMs are evaluated on such datasets after being edited by different techniques, it is hard to directly adopt the performance to assess the effectiveness of different knowledge editing methods in correcting hallucinations. Thus, the fundamental question remains insufficiently validated: ***Can knowledge editing really correct hallucinations in LLMs?*** We proposed HalluEditBench to holistically benchmark knowledge editing methods in correcting real-world hallucinations. First, we rigorously construct a massive hallucination dataset with 9 domains, 26 topics and more than $6,000$ hallucinations. Then, we assess the performance of knowledge editing methods in a holistic way on five dimensions including *Efficacy*, *Generalization*, *Portability*, *Locality*, and *Robustness*. Through HalluEditBench, we have provided new insights into the potentials and limitations of different knowledge editing methods in correcting hallucinations, which could inspire future improvements and facilitate progress in the field of knowledge editing.

## 1 INTRODUCTION

Large Language Models (LLMs) have shown superior performance in various tasks (Zhao et al., 2023). However, one critical weakness is that they may output hallucinations, referring to the non-factual information in generated content, for reasons such as the limit of models' internal knowledge scope or fast-changing world facts (Zhang et al., 2023). Considering the high cost of retraining LLMs from scratch, knowledge editing has been designed as a new paradigm to correct erroneous or outdated factual knowledge in LLMs (Wang et al., 2023c).

| Method | WikiData$_{recent}$ | ZsRE | WikiBio |
|---|---|---|---|
| Pre-edit | 47.40 | 37.49 | 61.35 |
| Post-edit (ROME) | 97.37 | 96.86 | 95.91 |
| Post-edit (MEMIT) | 97.10 | 95.86 | 94.68 |
| Post-edit (FT-L) | 56.30 | 53.82 | 66.70 |
| Post-edit (FT-M) | 100.00 | 99.98 | 100.00 |
| Post-edit (LoRA) | 100.00 | 100.00 | 100.00 |

Table 1: Performance measured by **Accuracy (%)** of Llama2-7B before editing ("Pre-edit") and after applying typical knowledge editing methods ("Post-edit") on common existing evaluation datasets.

Although there are many existing question-answering datasets such as WikiData$_{recent}$ (Cohen et al., 2024), ZsRE (Yao et al., 2023), and WikiBio (Hartvigsen et al., 2024) widely used for the evaluation of knowledge editing, one common issue is that they do not verify whether LLMs, before applying knowledge editing, actually generate hallucinated answers to the evaluation questions. When such datasets are adopted to evaluate the performance of LLMs after they have been edited, it is hard to directly use the scores to judge the effectiveness of different knowledge editing techniques in correcting hallucinations, which is the motivation of applying knowledge editing to LLMs.

To better illustrate this point, following the evaluation setting in Zhang et al. (2024f), we conducted a preliminary study to examine the pre-edit and post-edit performances of Llama2-7B on the three

---

[*]Equal Contribution. [†]Corresponding author.

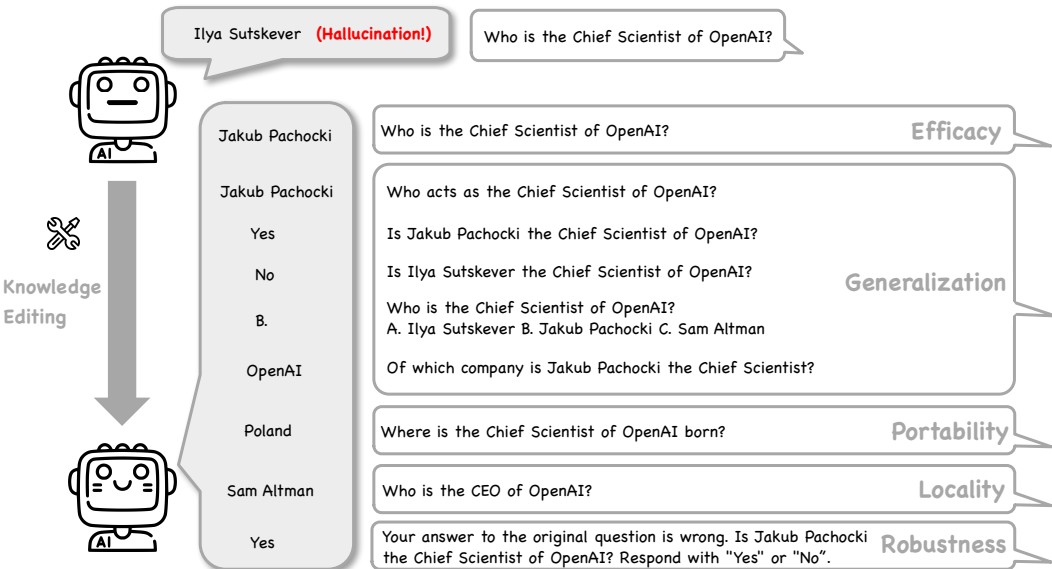

Figure 1: **Framework of HalluEditBench**. For real-world hallucinations, we holistically assess the performance of knowledge editing on *Efficacy*, *Generalization*, *Portability*, *Locality*, and *Robustness*.

aforementioned evaluation datasets. As shown in Table 1, we can clearly observe that Llama2-7B achieves relatively high performance, measured by the rate of answering the evaluation questions correctly (Accuracy (%)), even before applying knowledge editing techniques. Although the knowledge editing methods can bring an increase in accuracy, the high post-edit performance on these datasets cannot faithfully reflect the true effectiveness in correcting real-world hallucinations and may cause a distorted assessment. Thus, the fundamental question remains insufficiently validated: *Can knowledge editing really correct hallucinations in LLMs?*

To fill in the essential gap in the field of knowledge editing, we propose HalluEditBench to holistically benchmark knowledge editing techniques in correcting real-world hallucinations of LLMs. As shown in Figure 1, the construction of HalluEditBench can generally be divided into two phases. In the first phase, we constructed a massive hallucination dataset encompassing 9 domains and 26 topics based on Wikidata. For each of Llama2-7B, Llama3-8B, and Mistral-v0.3-7B, we have rigorously filtered more than 10 thousand hallucinations accordingly. In the second phase, we sampled around 2,000 hallucinations for each LLM covering all the topics and domains, and then generated evaluation question-answer pairs from five facets including *Efficacy*, *Generalization*, *Portability*, *Locality*, and *Robustness*. Through extensive empirical investigation on performance of 7 typical knowledge editing techniques, including FT-L (Zhu et al., 2020; Meng et al., 2022), FT-M (Zhang et al., 2024f), MEMIT (Meng et al., 2023), ROME (Meng et al., 2022), LoRA (Hu et al., 2022), ICE (Zheng et al., 2023), and GRACE (Hartvigsen et al., 2024), regarding the aforementioned five dimensions, we have provided novel insights into their potentials and limitations. A summary of the insights is as follows:

- **The effectiveness of knowledge editing methods in correcting real-world hallucinations could be far from what their performance on existing datasets suggests**, reflecting the potential unreliability of previous assessment of different knowledge editing techniques. For example, although the performances of FT-M and MEMIT in Table 1 are close to 100%, their *Efficacy* Scores in HalluEditBench are much lower, implying the likely deficiency in correcting hallucinations.

- **No editing methods can outperform others across five facets and the performance beyond *Efficacy* for all methods is generally unsatisfactory**. Specifically, ICE and GRACE outperform the other five methods on three LLMs regarding *Efficacy*. All editing methods except ICE only slightly improve or negatively impact the *Generalization* performance. Editing techniques except ICE could even underperform pre-edit LLMs on *Portability*. FT-M and ICE surpass others on *Locality* performance. ICE has a poor *Robustness* performance compared to other methods.

- **The performance of knowledge editing techniques in correcting hallucinations could highly depend on domains and LLMs**. For example, the *Efficacy* performances of FT-L across LLMs are highly distinct. Domains have a large impact on the *Locality* performance of ICE.

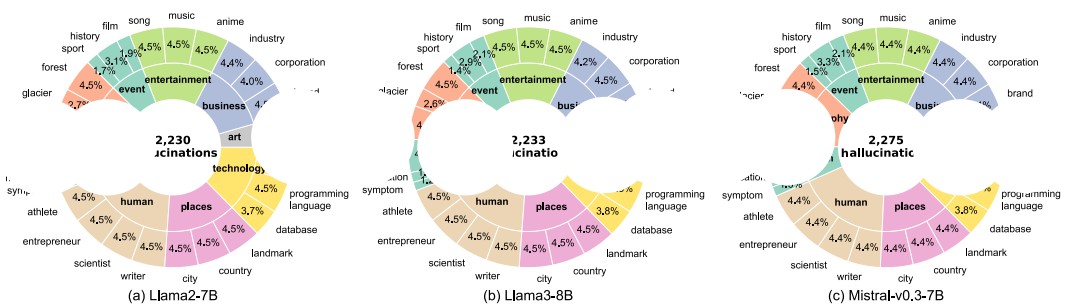

Figure 2: **Statistics of HalluEditBench Across Topics and Domains**.

# 2 HalluEditBench: HOLISTICALLY BENCHMARKING KNOWLEDGE EDITING METHODS IN CORRECTING REAL-WORLD HALLUCINATIONS

In this section, we will introduce the details of HalluEditBench, including the construction of the massive LLM hallucination dataset, the generation of evaluation question-answering pairs from five dimensions, evaluation metrics and the benchmarked knowledge editing techniques.

## 2.1 HALLUCINATION DATASET CONSTRUCTION

The goal of knowledge editing can generally be defined as transforming existing factual knowledge in the form of a knowledge triplet (subject $s$, relation $r$, object $o$) into a new one (subject $s$, relation $r$, object $o^*$). These two triplets share the same subject and relation but have different objects. A knowledge editing operation can be represented as $e = (s, r, o, o^*)$. Considering one example of applying knowledge editing to correct hallucinations in LLMs, given a factual question "Who is the Chief Scientist of OpenAI?", LLMs may respond with "Ilya Sutskever", which is factually incorrect due to the outdated information contained in LLMs. The editing operation can be $e = (s = $ OpenAI, $r = $ Chief Scientist, $o = $ Ilya Sutskever, $o^* = $ Jakub Pachocki). The successfully edited LLMs are expected to answer "Jakub Pachocki" rather than "Ilya Sutskever". Thus, we need to collect a large scale of knowledge triplets and factual questions to filter hallucinations.

Following existing editing datasets (*e.g.*, WikiData$_{recent}$ (Cohen et al., 2024) and WikiBio (Hartvigsen et al., 2024)), we also choose Wikidata as the factual knowledge source. In the *first* step, we retrieved $143,557$ raw knowledge triplets using the Wikidata Query Service (Query date: September 8th, 2024) from 26 topics, which can be categorized into 9 domains including *art*, *business*, *entertainment*, *event*, *geography*, *health*, *human*, *places*, and *technology*. Each topic has at least 100 triplets. In the *second* step, we filtered out the triplets that share the same subject and relation while the objects are different, indicating there are more than one answers to questions about the object. When we construct factual questions and compare LLM-generated answers with the objects of these triplets, it would be difficult to determine whether LLMs actually hallucinate the questions. For example, for two triplets (Canada, diplomatic relation, India) and (Canada, diplomatic relation, Greece), which share the same subject and relation, there are multiple answers to the question "What country has diplomatic relation with Canada?" In the *third* step, following Wang et al. (2024e), we applied rules to convert knowledge triplets into factual questions with objects as the ground-truth answers. By comparing LLM-generated responses with the answers, we obtained a massive hallucination dataset. Specifically, we collected $12,619$, $13,210$, and $14,366$ hallucinations for Llama2-7B, Llama3-8B, and Mistral-v0.3-7B respectively. Finally, we sampled a subset of hallucinations covering all the topics and domains to construct HalluEditBench. The distribution statistics are shown in Figure 2.

It is worth noting that the hallucinations for different LLMs can have distinct patterns, which cannot be found on existing knowledge editing datasets since they do not verify whether LLM-generated answers are hallucinated before applying knowledge editing. **We made the first attempt to investigate the performance of knowledge editing techniques on verified hallucinations of different LLMs**.

## 2.2 EVALUATION QA PAIR GENERATION AND METRICS

After constructing the hallucination dataset, we propose to holistically assess the performance of knowledge editing methods in correcting hallucinations from five facets including *Efficacy*, *Generalization*, *Portability*, *Locality*, and *Robustness*. First, we leveraged GPT-4o to generate evaluation

question-answering pairs for each facet based on the hallucination dataset as well as the factual verification questions in Section 2.1. Then we also manually inspect their quality. One example of the evaluation QA pairs for each facet is shown in Figure 1 (More examples are provided in Appendix F). The specific prompt design for GPT-4o is shown in Appendix A.

Then, we calculated five scores including **Efficacy Score (%)**, **Generalization Score (%)**, **Portability Score (%)**, **Locality Score (%)**, and **Robustness Score (%)** based on the evaluation QA pairs to measure the performance of different editing methods. Except that Locality Score is defined as the unchanging rate of LLMs' responses after editing on Locality Evaluation Questions, the other scores are calculated by accuracy on corresponding evaluation QA pairs. More details are as follows:

**Facet 1: Efficacy**    Efficacy Evaluation Questions are the same as the factual verification questions in the hallucination collection to ensure the pre-edit performance is $0\%$ regarding Efficacy Score. Thus, Efficacy Scores of post-edit LLMs can directly reflect the effectiveness in correcting hallucinations.

**Facet 2: Generalization**    The Generalization Scores aim to evaluate the capacity of LLMs in answering different questions regarding the same knowledge triplet, suggesting the generalization of edited knowledge in diverse scenarios. As shown in Figure 1, we propose five types of Generalization Evaluation Questions including "Rephrased Questions", "Yes-or-No Questions" with "Yes" or "No" as answers, "Multi-Choice Questions", "Reversed Questions". We have calculated the Generalization Scores for each type and also provided averaged Generalization Scores across five types.

**Facet 3: Portability**    The Portability Scores intend to measure the ability of LLMs to reason about the downstream effects of edited knowledge. Thus, we design the Efficacy Evaluation Questions with $N$ hops ($N = 1 \sim 6$) as Portability Evaluation Questions. When $N = 2$, the example is shown in Figure 1. When the answer to the question "Who is the Chief Scientist of OpenAI?" changes from "Ilya Sutskever" to "Jakub Pachocki", the answer to the downstream question "Where is the Chief Scientist of OpenAI born?" should also change from "Russia" to "Poland".

**Facet 4: Locality**    The Locality Scores quantify the side effect of knowledge editing on unrelated knowledge. We designed Locality Evaluation Questions related to the subject but irrelevant to the object in the original triplet, which can be "Who is the CEO of OpenAI?" for the aforementioned example. Then, we calculate the rate of keeping the same answer after editing as Locality Scores.

**Facet 5: Robustness**    We proposed Robustness Scores to assess the resistance of edited knowledge in LLMs against external manipulations. Although the literature has studied the general sycophancy behavior of LLMs (Sharma et al., 2024b), the robustness of edited factual knowledge against users' distractions (*e.g.*, "Your answer to the original question is wrong.") is under-explored. After post-edit LLMs are tested with Efficacy Evaluation Questions, we further prompted them with Robustness Evaluation Questions, which are exemplified in Figure 1, for $M$ turns ($M = 1 \sim 10$) and calculated the rate of "Yes" for each round as the Robustness Scores, reflecting the extent to which LLMs insist on the corrected knowledge. Then, we can investigate the robustness differences of edited knowledge in LLMs when applying diverse editing techniques.

## 2.3 KNOWLEDGE EDITING TECHNIQUES

We propose to categorize the majority of existing knowledge editing techniques into the following 4 types and chose 7 representative techniques (more details are in Appendix B) in HalluEditBench.

- **Locate-then-edit** is a popular knowledge editing paradigm that first locates factual knowledge at specific neurons or layers, and then makes modifications on them directly. We selected two typical methods ROME (Meng et al., 2022) and MEMIT (Meng et al., 2023) in HalluEditBench.

- **Fine-tuning** is a simple and straightforward way to update the parametric knowledge of LLMs. We selected three variations FT-L (Meng et al., 2022), FT-M (Zhang et al., 2024f), and LoRA (Hu et al., 2022), which mitigate the catastrophic forgetting and overfitting issues of standard fine-tuning.

- **In-Context Editing** is a training-free paradigm that associates LLMs with in-context knowledge directly (Zheng et al., 2023; Shi et al., 2024; Fei et al., 2024). We adopted a simple baseline ICE method in Zheng et al. (2023) that puts the new fact in context and does not require demonstrations.

- **Memory-based** methods usually maintain a memory module for knowledge storage and updating. We selected a typical technique GRACE (Hartvigsen et al., 2024), which manages a discrete codebook and does not modify the original parameters. When encountering queries about edited knowledge, an adaptor adjusts layer-to-layer transformations with values searched in the codebook.

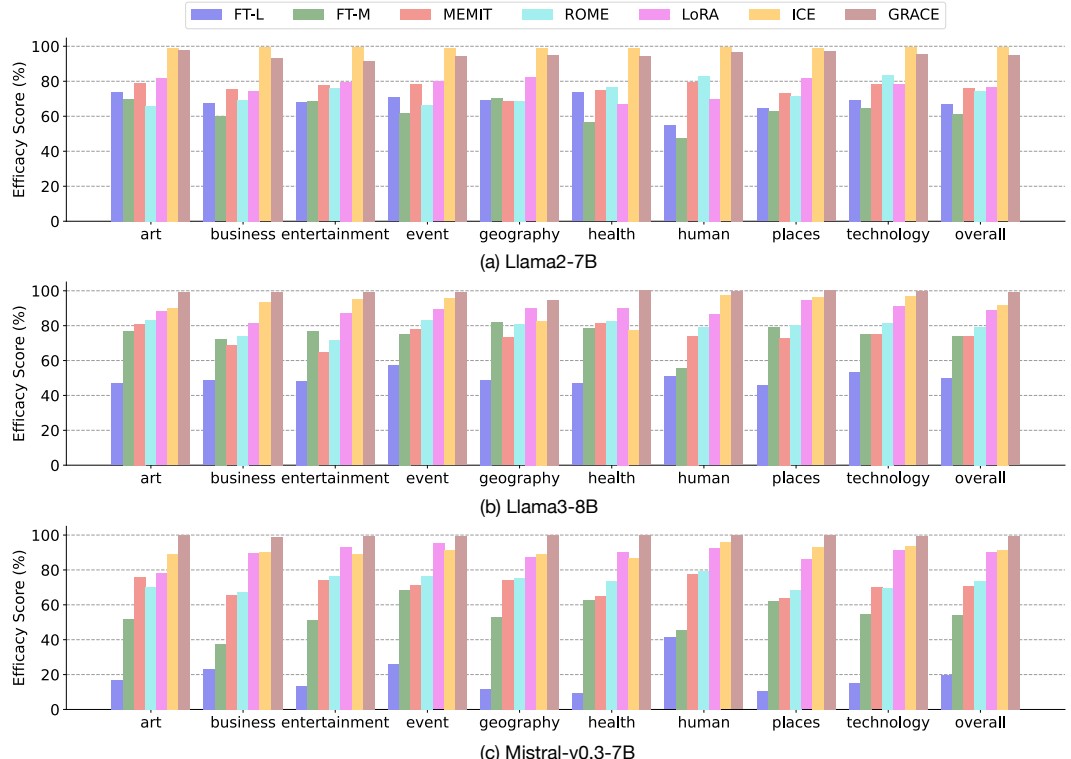

Figure 3: **Efficacy Scores of Knowledge Editing Methods**. The "overall" refers to the Efficacy Score (%) on the whole HalluEditBench embracing 9 domains for different methods. The Efficacy Score on each domain is also reported. Efficacy scores (%) are measured by the accuracy on Efficacy Evaluation Question-answer Pairs, where the pre-edit scores of each LLM are ensured 0%.

## 3 RESULTS AND ANALYSIS

In this section, we comprehensively analyze the experiment results on 9 domains and the overall performance on the whole HalluEditBench for different knowledge editing techniques from five facets including *Efficacy*, *Generalization*, *Portability*, *Locality*, and *Robustness*.

### 3.1 FACET 1: EFFICACY

Since we have ensured that LLMs generate hallucinated answers to the Efficacy Evaluation Questions before editing, the pre-edit Efficacy Score for all editing techniques is 0%. Thus, Efficacy Scores in Figure 3 can directly reflect the effectiveness of different techniques in correcting real-world hallucinations. We find that **the effectiveness of some techniques can be far from what their performance on previous datasets suggests**, implying the potential unreliability of their previous evaluation. For example, as shown in Table 1, although FT-M achieves near 100% performance in existing datasets such as WikiData$_{recent}$, ZsRE, and WikiBio, its overall Efficacy Scores on Llama2-7B and Mistral-v0.3-7B are only around 60%. There is a similar performance drop for MEMIT.

Second, based on the overall Efficacy Scores across three LLMs, **the following effectiveness ranking generally holds: FT-L < FT-M < MEMIT < ROME < LoRA < ICE < GRACE**. We can observe that ICE and GRACE, which both preserve original weights in LLMs, outperform the other methods, implying **the potential disadvantage of directly modifying parameters for knowledge editing**.

Third, we notice that **efficacy scores of knowledge editing techniques could highly depend on domains and LLMs**. For example, the scores of FT-L on different domains and LLMs could be highly distinct. The performance of FT-L and FT-M on Llama3-8B is higher than that on Mistral-v0.3-7B.

> **Insight 1:** (1) The current assessment of knowledge editing could be unreliable; (2) ICE and GRACE outperform parameter-modifying editing techniques such as fine-tuning and "Locate-then-Edit" methods on *Efficacy*; (3) Domains and LLMs could have a high impact on *Efficacy*.

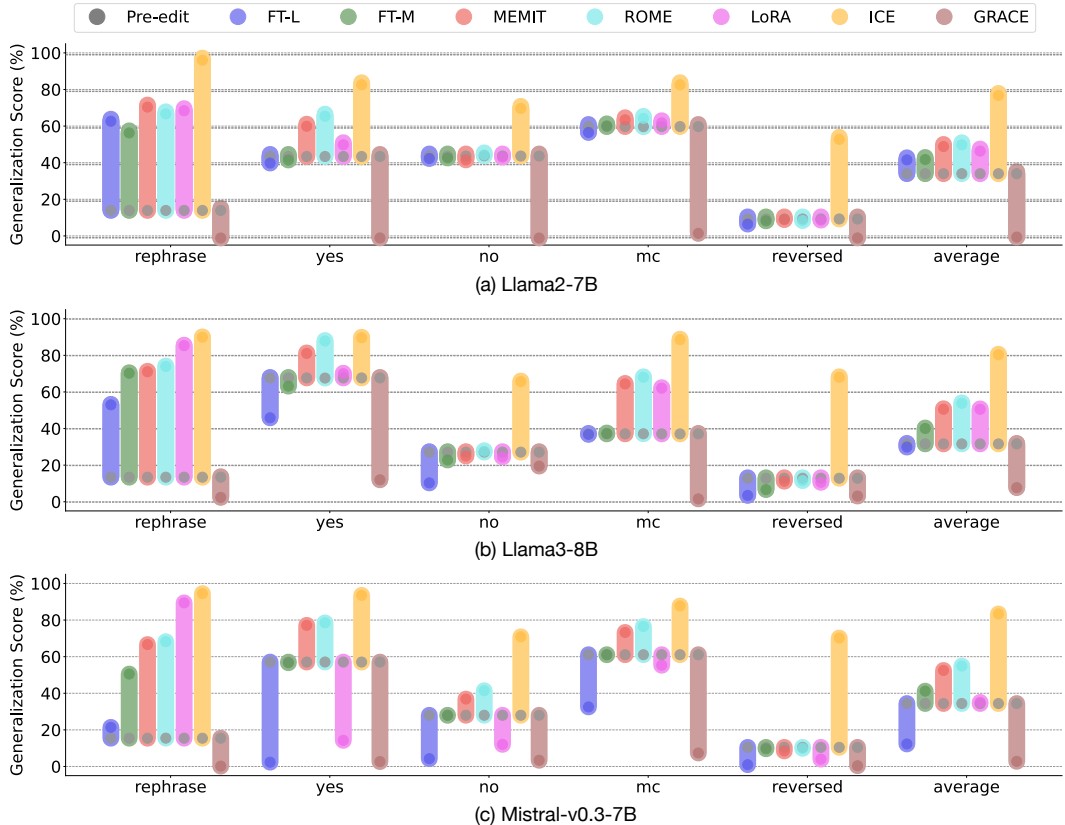

Figure 4: **Generalization Scores of Knowledge Editing Methods**. Generalization Scores (%) are measured by accuracy on five types of Generalization Evaluation Questions including Rephrased Questions ("rephrase"), Yes-or-No Questions with "Yes" or "No" as answers ("yes" or "no"), Multi-Choice Questions ("mc"), Reversed Questions ("reversed"). The "average" refers to averaged scores over five question types. The figure only shows the overall Generalization Scores for each type on the whole HalluEditBench. Generalization Scores for each domain are given in Appendix E.1.

## 3.2 FACET 2: GENERALIZATION

As shown in Figure 4, even though the pre-edit Efficacy Score performances for different editing techniques on three LLMs are ensured $0\%$, it is worth noting that the pre-edit Generalization Score performance is not $0\%$ for each question type, illustrating that **the manifestation of hallucination actually depends on the design of question prompts**. Given a group of diverse question prompts for the same knowledge triplet, LLMs may hallucinate some questions but answer others correctly.

Surprisingly, we find that **post-edit Generalization Scores could even be lower than pre-edit scores** for the same LLM and question type, demonstrating the potential negative effect caused by knowledge editing. In more detail, we can observe a clear performance drop for GRACE across all the question types, and for FT-L and LoRA on some question types.

Comparing the ranking of Efficacy Scores in Figure 3 with Figure 4, we can explicitly see that **higher Efficacy Scores do not also necessarily indicate higher Generalization Scores**. Especially, although GRACE almost surpasses all the other editing techniques regarding Efficacy Scores, it largely degrades the Generalization Scores compared to pre-edit performance. In addition, **all editing methods except ICE only slightly improve or even hurt Generalization Scores**.

> **Insight 2:** (1) The manifestation of hallucination depends on question design; (2) Higher *Efficacy* Scores do not also necessarily indicate higher *Generalization* Scores; (3) All editing techniques except ICE only slightly improve or negatively impact the *Generalization* performance.

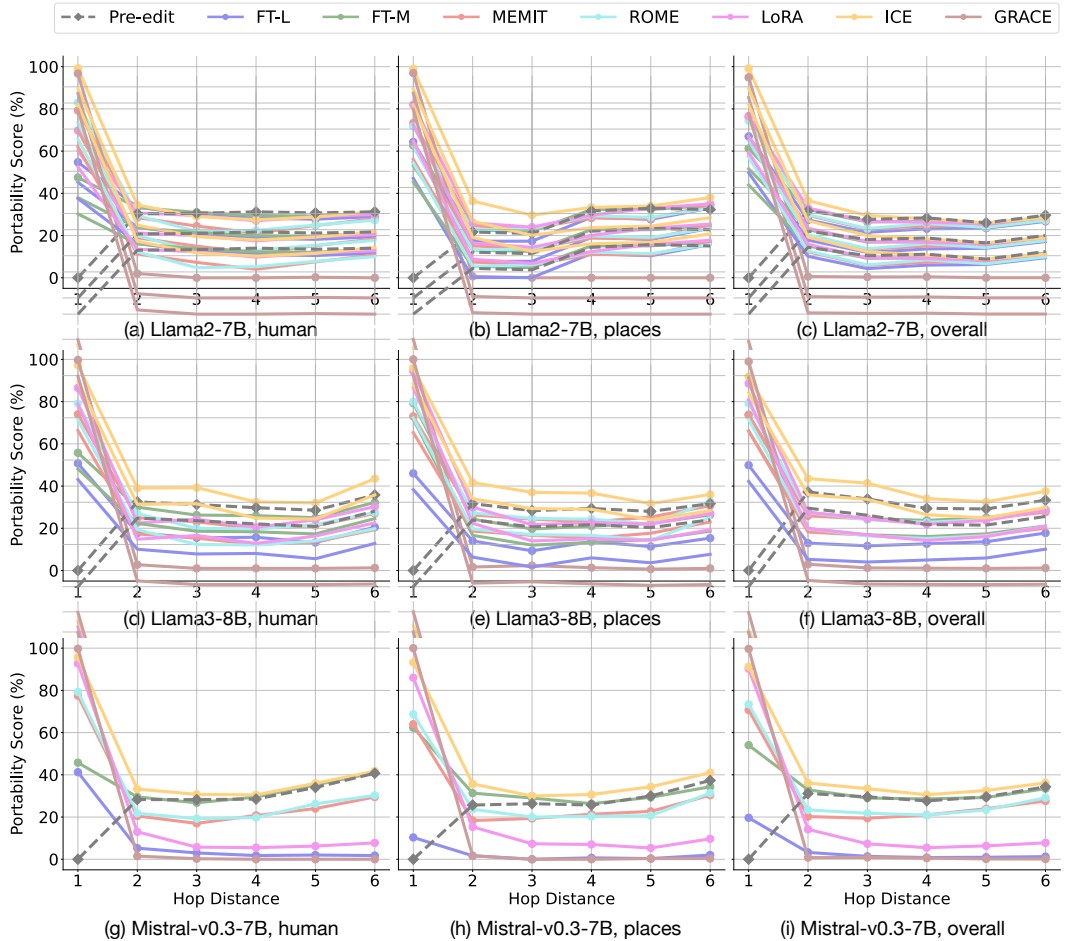

Figure 5: **Portability Scores of Knowledge Editing Methods**. Portability Scores (%) are measured by the accuracy on Portability Evaluation Questions, which are Efficacy Evaluation Questions with $N$ hops ($N = 1 \sim 6$). The Portability Evaluation Questions are the same as Efficacy Evaluation Questions when $N$ is 1. The Portability Scores on two domains "human" and "places" are reported in the figure. The results for more domains are given in Appendix E.2. The "overall" refers to the Portability Score (%) on the whole HalluEditBench embracing 9 domains.

### 3.3 FACET 3: PORTABILITY

Figure 5 demonstrates the pre-edit and post-edit Portability Scores for Portability Evaluation Questions with $N$ hops ($N = 1 \sim 6$). When $N = 1$, the Portability Evaluation Questions are the same as Efficacy Evaluation Questions, suggesting that the Portability Scores are 0. Similar to Figure 4, we discover that the pre-edit Portability Scores are not zero for $2 \sim 6$ hops, indicating **LLMs do not necessarily need to reason based on single-hop knowledge to answer multi-hop questions**. We hypothesize that this is because LLMs may directly memorize the answers to multi-hop questions.

We surprisingly find that except that ICE may bring marginal improvement to the pre-edit performance, **the other knowledge editing techniques even mostly underperform pre-edit Portability Scores**, showing another type of negative effect of knowledge editing and **LLMs may not really reason with the edited knowledge in multi-hop questions** for most knowledge editing methods. Comparing single-hop and multi-hop performance, we observe a sharp decrease for all the editing methods, which further underscores **the challenges of answering multi-hop questions with edited knowledge**.

> **Insight 3:** (1) LLMs may memorize answers rather than reason based on single-hop knowledge for multi-hop questions; (2) Editing methods marginally improve or degrade pre-edit *Portability* Scores, implying LLMs may not really reason with edited knowledge in multi-hop questions.

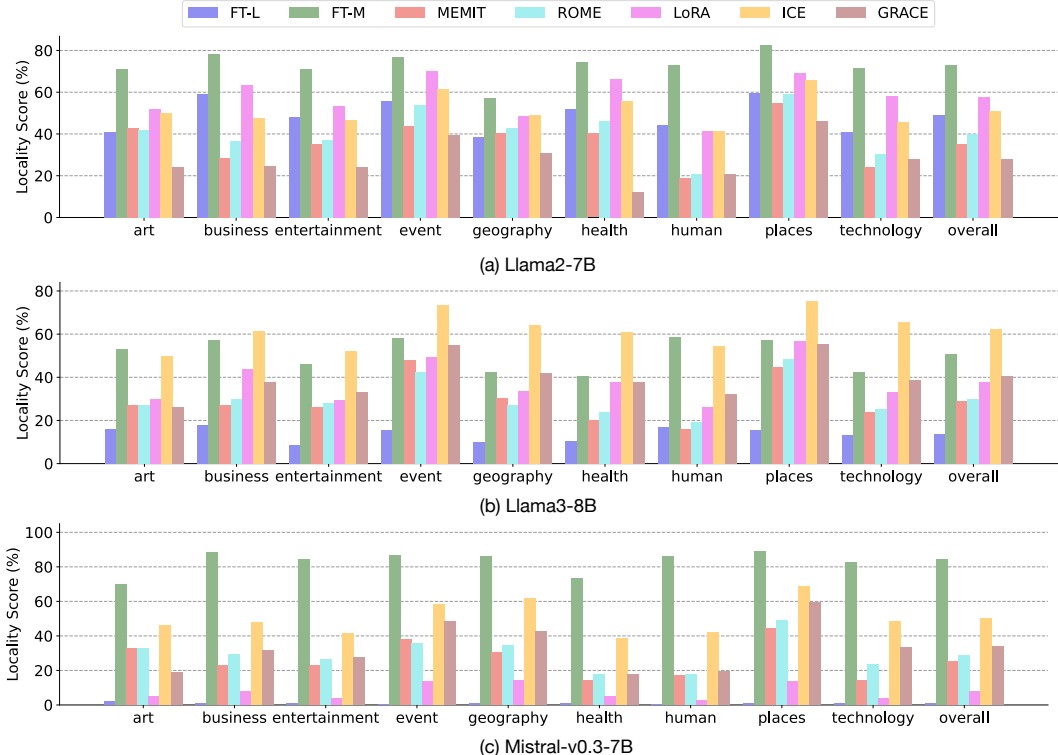

Figure 6: **Locality Scores of Knowledge Editing Methods**. Locality Scores (%) are measured by the unchanging rate on Locality Evaluation Questions after applying knowledge editing methods on LLMs. A higher Locality Score indicates that there is a higher percentage of LLMs' answers to the unrelated questions keeping the same and a less side effect on general knowledge in LLMs. The "overall" refers to the Locality Score (%) on the whole HalluEditBench embracing 9 domains for different methods. The Locality Score on each domain is also reported in the figure.

## 3.4 FACET 4: LOCALITY

Figure 6 shows the Locality Scores of different editing techniques in each domain and the whole HalluEditBench, reflecting the side effect of knowledge editing on unrelated knowledge encoded in LLMs. Based on the overall Locality Scores, we can observe that **the performance of all editing methods except FT-M and ICE is unsatisfactory**. In particular, the overall Locality Scores for all editing techniques except FT-M and ICE on Llama3-8B and Mistral-v0.3-7B are below $40\%$, suggesting a high undesired impact on LLMs' answers to unrelated factual questions, though FT-M achieves an overall score of around $80\%$ on Mistral-v0.3-7B and ICE gains $60\%$ on Llama3-8B.

Furthermore, we notice that **domains and LLMs have a high impact on the Locality Scores of knowledge editing methods**. For example, the Locality Score for ICE in the "places" domain in Llama3-8B is near $80\%$, while the performance drops to only about $50\%$ in the "art" domain for the same LLM. Although FT-L obtains a Locality Score around $60\%$ in the "business" domain on Llama2-7B, its performance in the same domain on Mistral-v0.3-7B is almost $0\%$.

Due to the impact of LLMs, we observe that **the rankings by Locality Scores for editing techniques on different LLMs are highly distinct**. For example, the Locality ranking on Llama2-7B is GRACE < MEMIT < ROME < FT-L < ICE < LoRA < FT-M. However, the ranking changes to FT-L < LoRA < MEMIT < ROME < GRACE < ICE < FT-M on Mistral-v0.3-7B. Comparing Figure 3 with Figure 6, we find **there is no noticeable correlation between Efficacy and Locality for different editing techniques**. FT-M achieves relatively high Locality Scores despite its low Efficacy Scores.

**Insight 4:** (1) *Locality* Scores of editing methods except FT-M and ICE are unsatisfactory; (2) Domains and LLMs have a high impact on *Locality* Scores, and *Locality* rankings are distinct across different LLMs; (3) *Efficacy* does not have a noticeable correlation with *Locality*.

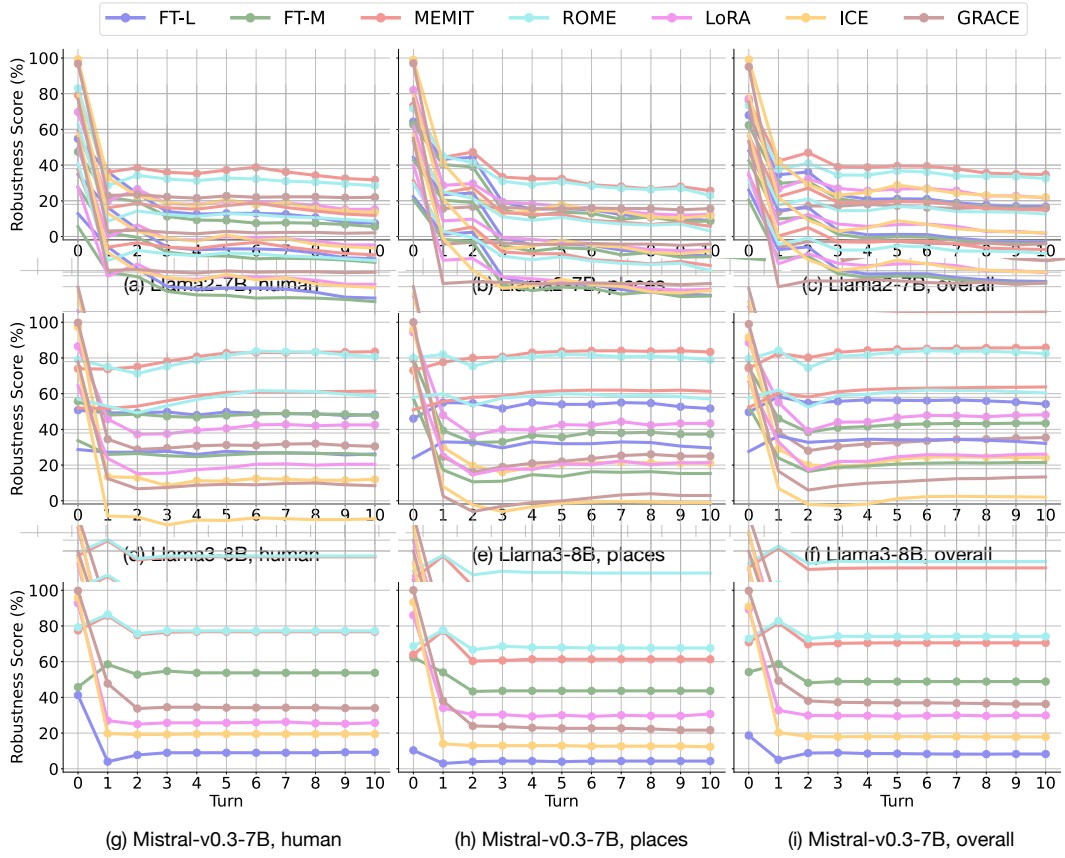

Figure 7: **Robustness Scores of Knowledge Editing Methods**. Robustness Scores are calculated by the accuracy on Robustness Evaluation Questions with $M$ turns ($M = 1 \sim 10$). We regard Efficacy Scores as the Robustness Scores when $M$ is 0. The Robustness Scores on two domains "human" and "places" are reported in the figure. The results for more domains are given in Appendix E.3. The "overall" refers to the Robustness Score (%) on the whole HalluEditBench embracing 9 domains.

## 3.5 FACET 5: ROBUSTNESS

We proposed Robustness Scores (%) to evaluate the resistance of edited knowledge against distractions in prompts. Initially ($M = 0$), LLMs are assessed with Efficacy Evaluation Questions. Then ($M = 1 \sim 10$), LLMs are sequentially prompted with Robutness Evaluation Questions, which are exemplified in Figure 1, for $M$ turns. Robustness Scores are calculated with the percentage of "Yes" in each round. A higher Robustness Score indicates that there is a larger percentage of LLMs can resist external manipulations in the prompt and a higher extent of robustness for the edited knowledge.

First, based on overall Robustness Scores, we observe that **LLMs themselves have a large impact on the robustness of edited knowledge**. **The same editing method could show distinct trends as turns increase on different LLMs**. For example, all editing methods have a sharp drop when turns go up on Llama2-7B, showing a low level of robustness. However, MEMIT, ROME on Llama3-8B and Mistral-v0.3-7B maintain almost the same and relatively high performance as turns increase, suggesting a comparatively high level of robustness for the edited knowledge.

Then, we notice that **both ICE and GRACE have a low level of robustness** though they outperform the other five editing techniques regarding Efficacy Scores, showing **the potential weaknesses on robustness of parameter-preserving knowledge editing methods**. However, parameter-modifying editing techniques do not necessarily have high robustness, which is exemplified by LoRA.

**Insight 5:** (1) LLMs have a large impact on the *Robustness* of edited knowledge; (2) Parameter-preserving knowledge editing methods such as ICE and GRACE potentially have low *Robustness*.

## 4 RELATED WORK

Knowledge editing techniques have attracted increasing attention for their efficiency advantages in addressing obsolete or hallucinated information in LLMs (Wang et al., 2023c; Zhang et al., 2024f). In general, the existing editing techniques can be categorized into four types including *Locate-then-edit* (Meng et al., 2022; 2023), *Fine-tuning based* (Gangadhar & Stratos, 2024; Zhu et al., 2020; Wang et al., 2024a), *In-Context Editing* (Zheng et al., 2023; Shi et al., 2024; Fei et al., 2024), and *Memory-based* (Wang et al., 2024d; Hartvigsen et al., 2024; Mitchell et al., 2022; Yu et al., 2023). Recently, many benchmarks have been built to investigate the properties of knowledge editing from different perspectives (Rosati et al., 2024; Wu et al., 2023; Ge et al., 2024a; Ma et al., 2023; Wei et al., 2023; 2024a; Zhong et al., 2023; Lin et al., 2024; Huang et al., 2024c; Liu et al., 2024c; Akyürek et al., 2023; Li et al., 2024a;f; 2023b; Gu et al., 2024; Powell et al., 2024; Yang et al., 2025; Du. et al., 2025; Zhang et al., 2024a). For example, Gu et al. (2024) proposed a benchmark to assess the side effect of 4 popular editing methods on 3 LLMs across 8 general capacity tasks. Rosati et al. (2024) built a new evaluation protocol to measure the efficacy and impact of knowledge editing in long-form generation. Wei et al. (2024a) introduced a multilingual knowledge editing benchmark embracing five languages. However, considering the fundamental motivation of applying knowledge editing to LLMs, which is to correct hallucinations, there is a pressing need to build a real-world hallucination dataset with rigorous verification and systematically analyze the performance of different editing methods. Thus, we proposed HalluEditBench to fill in the gap and provided new insights to facilitate the progress in the field of knowledge editing.

## 5 CONCLUSION

In this paper, we have built a new benchmark HalluEditBench to holistically assess diverse knowledge editing techniques in correcting real-world hallucinations. First, we meticulously construct a massive and comprehensive hallucination dataset based on Wikidata with 9 domains, 26 topics, and more than $6,000$ hallucinations. Then, we systematically investigate the performance of different knowledge editing methods from five perspectives including *Efficacy*, *Generalization*, *Portability*, *Locality*, and *Robustness*. Our findings reveal that previous benchmarks cannot reflect the true effectiveness of knowledge editing methods in correcting real-world hallucinations and current editing methods mostly show limited performance across five dimensions. We also offer valuable and actionable insights to inspire future advancements in knowledge editing for large language models.

## ACKNOWLEDGMENTS

This material is based upon work supported by the U.S. Department of Homeland Security under Grant Award Number 17STQAC00001-07-04, NSF awards (SaTC-2241068, IIS-2506643, and POSE-2346158), a Cisco Research Award, and a Microsoft Accelerate Foundation Models Research Award. The views and conclusions contained in this document are those of the authors and should not be interpreted as necessarily representing the official policies, either expressed or implied, of the U.S. Department of Homeland Security and the National Science Foundation.

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

# Content of Appendix

## A  REPRODUCIBILITY STATEMENT

We conduct the experiments on NVIDIA RTX A6000 GPUs. The decoding temperatures are 0 to ensure the reproducibility. The model checkpoints are downloaded from https://huggingface.co/. The specific download links are as follows:

- Llama2-7B: https://huggingface.co/meta-llama/Llama-2-7b-chat-hf
- Llama3-8B: https://huggingface.co/meta-llama/Meta-Llama-3-8B-Instruct
- Mistral-v0.3-7B: https://huggingface.co/mistralai/Mistral-7B-Instruct-v0.3

We adopt GPT-4o with the prompt below to generate *Generalization* and *Locality* evaluation questions:

---

Given a fact triplet (subject, relation, object), a question asking for the object, and a wrong answer, the correct answer to the question should be the object in the triplet.

Generate the following types of questions:
   1. Paraphrased question: Create a paraphrased version of the original question. The correct answer should still be the object from the triplet.
   2. Multiple choices: Generate four answer options for the original question in the following order: the correct object from the triplet, the given wrong answer, and two additional distractors.
   3. Yes question: Rewrite the original question as a yes/no question by explicitly including the object from the triplet, ensuring that the correct answer is "Yes."
   4. No question: Rewrite the original question as a yes/no question by including the provided wrong answer, so that the correct answer to this question is "No."
   5. Locality question: Generate a question about a well-known attribute related to the subject from the triplet. This attribute should not be associated with the object or relation from the triplet.
   6.  Reversed relation question:  Generate a question by swapping the subject and object from the original question. The answer should now be the subject from the triplet.

Output the result in JSON format with the following keys: "paraphrased_question", "multiple_choices", "yes_question", "no_question", "locality_question", and "reversed_relation_question."

---

We adopt GPT-4o with the following prompt to generate evaluation questions in *Portability* aspect.

---

Given a subject, a relation, a 1-hop question, and its answer, create 2-hop, 3-hop, 4-hop, 5-hop, and 6-hop questions, along with their correct answers.
Always use the provided subject and relation to create multi-hop questions and include the preceding question in the subsequent question (for example, include the 2-hop question in 3-hop question, include the 3-hop question in 4-hop question).
DO NOT include the correct answer to any previous multi-hop question in subsequent ones (for example, do not include the correct answer to the 2-hop question in the 3-hop or 4-hop questions).
Ensure that the answers for all multi-hop questions are accurate, and do not use 'N/A' as an answer.
You must include the given subject and relation in all of the 2-hop, 3-hop, 4-hop, 5-hop, and 6-hop questions. Output in JSON format. An example is provided below:

Example input:
subject: Amazon, relation: founder
1hop_question: Who is the Amazon founder? 1hop_answer: Jeff Bezos

Example output:
{
   "2hop_question": "Who is the spouse of the Amazon founder?",   "2hop_answer": "MacKenzie Scott",
   "3hop_question":   "Which university did the spouse of the Amazon founder attend for their undergraduate studies?",   "3hop_answer":   "Princeton University",
   "4hop_question":   "In which city is the university that the spouse of the Amazon founder attended located?",   "4hop_answer":   "Princeton",
   "5hop_question":   "In which state is the city located where the university that the spouse of the Amazon founder attended is situated?",   "5hop_answer":   "New Jersey",
   "6hop_question":   "In which country is the state located where the city is situated that contains the university the spouse of the Amazon founder attended?",   "6hop_answer":   "United States",
}

---

## B  DETAILS OF THE BENCHMARKED KNOWLEDGE EDITING TECHNIQUES

**FT-L** (Zhu et al., 2020; Meng et al., 2022) Constrained Fine-Tuning (FT-L) is a targeted approach to fine-tuning that focuses on adjusting a specific layer within a model's feed-forward network (FFN). Guided by causal tracing results from ROME, FT-L modifies the layer most associated with the desired changes. The goal of FT-L is to fine-tune the model by maximizing the likelihood of the target sequence, particularly focusing on the prediction of the last token, ensuring that the model adapts to modified facts without affecting its broader performance. To achieve this, explicit parameter-space norm constraints are applied to the weights, ensuring minimal interference with unmodified facts and preserving the integrity of the model's original knowledge.

**FT-M** (Zhang et al., 2024f) In contrast to FT-L, which fine-tunes by maximizing the probability of all tokens in the target sequence based on the last token's prediction, Fine-Tuning with Masking (FT-M) refines this approach to align more closely with the traditional fine-tuning objective. FT-M also targets the same FFN layer identified by causal tracing but employs a masked training strategy. Specifically, it uses cross-entropy loss on the target answer while masking out the original text, ensuring that the model is trained directly on the relevant target content. This approach mitigates potential deviations from the original fine-tuning objective and provides a more precise adjustment of the model's weights with minimal disruption to unrelated model behavior.

**LoRA** (Hu et al., 2022) Low-Rank Adaptation (LoRA) is a parameter-efficient fine-tuning method that enhances training efficiency by introducing trainable rank decomposition matrices into Transformer layers. Rather than updating the original model parameters directly, LoRA focuses on training expansion and reduction matrices with low intrinsic rank, which allows for significant dimensionality reduction and thus faster training. Specifically, LoRA freezes the pretrained model weights and optimizes rank decomposition matrices to indirectly adapt dense layers without altering the original parameters. This approach greatly reduces the number of trainable parameters needed for downstream tasks, enabling more efficient training and lowering hardware requirements.

**ROME** (Meng et al., 2022) Rank-One Model Editing (ROME) is a "Locate-then-Edit" technique designed to modify factual associations within transformer models. ROME localizes these associations along three key dimensions: (1) the MLP module parameters, (2) within a range of middle layers, and (3) specifically during the processing of the last token of the subject. It employs causal intervention to trace the causal effects of hidden state activations, identifying the specific modules that mediate the recall of factual information. Once these decisive MLP modules are localized, ROME makes small, targeted rank-one changes to the parameters of a single MLP module, effectively altering individual factual associations while minimizing disruption to the overall model behavior. This precise parameter adjustment enables direct updates to the model's factual knowledge.

**MEMIT** (Meng et al., 2023) Mass Editing Memory in a Transformer (MEMIT) builds upon ROME to generalize the editing of feedforward networks (FFNs) in pre-trained transformer models for mass knowledge updates. While ROME focuses on localizing and modifying factual associations within single layers, MEMIT extends this strategy to perform mass edits across a range of critical layers. MEMIT uses causal tracing to identify MLP layers that act as mediators of factual recall, similarly to ROME, but scales the process to enable the simultaneous insertion of thousands of new memories. By explicitly calculating parameter updates, MEMIT targets these critical layers and updates them efficiently, offering a scalable multi-layer update algorithm that enhances and expands upon ROME's capability to modify knowledge across many memories concurrently, achieving orders of magnitude greater scalability.

**ICE** (Zheng et al., 2023) In-Context Knowledge Editing (IKE) leverages in-context learning (ICL) to modify model outputs without altering the model's parameters. This approach reduces computational overhead and avoids potential side effects from parameter updates, offering a more efficient and safer way to modify knowledge in large language models. IKE enhances interpretability, providing a human-understandable method for calibrating model behaviors. It achieves this by constructing three types of demonstrations-copy, update, and retain-that guide the model in producing reliable fact editing through the use of a demonstration store. This store, built from training examples, allows the model to retrieve the most relevant demonstrations to inform its responses, improving accuracy in modifying specific factual outputs. In-Context Editing (ICE) is a simple baseline variant of IKE, which directly uses the new fact as context without additional demonstrations.

**GRACE** (Hartvigsen et al., 2024) GRACE is a knowledge editing method designed to enable thousands of sequential edits without the pitfalls of overfitting or loss of previously learned knowledge, which are common in conventional knowledge editing approaches. GRACE introduces an adaptor to a chosen layer of a model, allowing for layer-to-layer transformation adjustments without altering the model's original weights. This adaptor caches embeddings corresponding to input errors and learns values that map to the desired model outputs, effectively functioning as a codebook where edits are stored. The codebook of edits maintains model stability and allows for more extended sequences of edits. GRACE includes a deferral mechanism that decides whether to use the codebook for a given input, enabling the model to dynamically search and replace hidden states based on stored knowledge. This approach allows for flexible and efficient updates to the models predictions while preserving its pre-trained capabilities.

## C   A MORE DETAILED RELATED WORK

Knowledge Editing has been adopted as one of the mainstream paradigms to address the hallucinations in LLMs efficiently (Chen & Shu, 2024a; Tonmoy et al., 2024; Li et al., 2024e). Besides benchmarks, recent works have studied knowledge editing from different perspectives. The first line of works aims to probe into the relationship between localization and editing and gain a deeper understanding of the working mechanisms of different techniques (Wang et al., 2024b; Niu et al., 2024; Hase et al., 2024a;b; Ferrando et al., 2024; Gupta et al., 2024; Chen et al., 2024e;d; Zou et al., 2023; Yao et al., 2024; Wu et al., 2025). For example, Hase et al. (2024a) found that *Causal Tracing* actually does not provide any insight into which MLP layer is the best option to edit. The second line of works intends to enhance the performance and applicability of knowledge editing in specific scenarios (Rozner et al., 2024; Jiang et al., 2024a;b; Zhang et al., 2024d;c;e;b;g; 2025a;b; Wu et al., 2024; Qi et al., 2024; Sharma et al., 2024a; Li et al., 2024c;b; Fang et al., 2024; Wang & Li, 2024a;b; Wang et al., 2024g;f;d; 2023b; Cheng et al., 2024b;a; Xie et al., 2024; Bi et al., 2024c;b;a; Chen et al., 2024c;b; Wei et al., 2024b; Fei et al., 2024; Xu et al., 2024; Gu et al., 2023; Yin et al., 2024; Cai et al., 2024a; Liu et al., 2024b; 2025; Ge et al., 2024b; Deng et al., 2024; Peng et al., 2024; Zhao et al., 2025; Jiang et al., 2025; Li et al., 2025; Lu et al., 2024; Zeng et al., 2024; Gu et al., 2025). For example, Ma et al. (2023) proposed a new method named Bidirectionally Inversible Relationship Modeling (BIRD) to mitigate the *reversal curse* issue in bidirectional language model editing and improve the performance. The third line of works investigates the side effect of knowledge editing techniques (Hsueh et al., 2024; Gu et al., 2024; Hoelscher-Obermaier et al., 2023; Hua et al., 2024; Yang et al., 2024a;b; Li et al., 2023a; Cohen et al., 2024). For example, Yang et al. (2024a) discovered that even one single edit could cause a significant performance degradation in mainstream benchmarks. The fourth line of works explores the potential misuse risks of knowledge editing or its applications beyond correcting hallucinations (Chen et al., 2024a; Uppaal et al., 2024; Wang et al., 2024c; Cai et al., 2024b; Yan et al., 2024; Zhang et al., 2025c; Grimes et al., 2024; Li et al., 2024d; Youssef et al., 2024; 2025). For example, Chen et al. (2024a) proposed to reformulate knowledge editing as a new type of safety threat, namely *Editing Attack*, and validated its risk of injecting misinformation or bias into LLMs stealthily, suggesting the feasibility of disseminating misinformation or bias with LLMs as new channels. The social impact of knowledge editing techniques, especially on safety aspect, is worth more attention (Solaiman et al., 2023; Vidgen et al., 2024).

## D   IMPACT STATEMENT

Misinformation is a longstanding threat for online safety and public trust (Chen et al., 2022; Wang et al., 2023a). The conventional countermeasures include *detection* (Shu et al., 2017; Nan et al., 2024; 2023; Liu et al., 2024a), *intervention* (Bak-Coleman et al., 2022; Aghajari et al., 2023; Hartwig et al., 2024; Yue et al., 2024; He et al., 2023) and *attribution* (Huang et al., 2024a;b; Beigi et al., 2024). Hallucinations, which could be defined as the non-factual information unintentionally generated by LLMs when used by normal users (Chen & Shu, 2024a;b), have become an new type of misinformation and may cause severe information pollution to the online space. Besides methods such as Retrieval-Augmented Generation (Shi et al., 2025; Ni et al., 2025; Zhou et al., 2024), knowledge editing is a promising paradigm to correct hallucinations and contribute to the fight against the misinformation crisis in the era of LLMs, due to its advantage of avoiding retraining from scratch. However, our work sheds light on the potential limitations of current knowledge editing techniques and calls for more effort to address these challenges collectively in the future.

# E MORE EXPERIMENT RESULTS

## E.1 GENERALIZATION SCORES OF KNOWLEDGE EDITING METHODS ON EACH DOMAIN

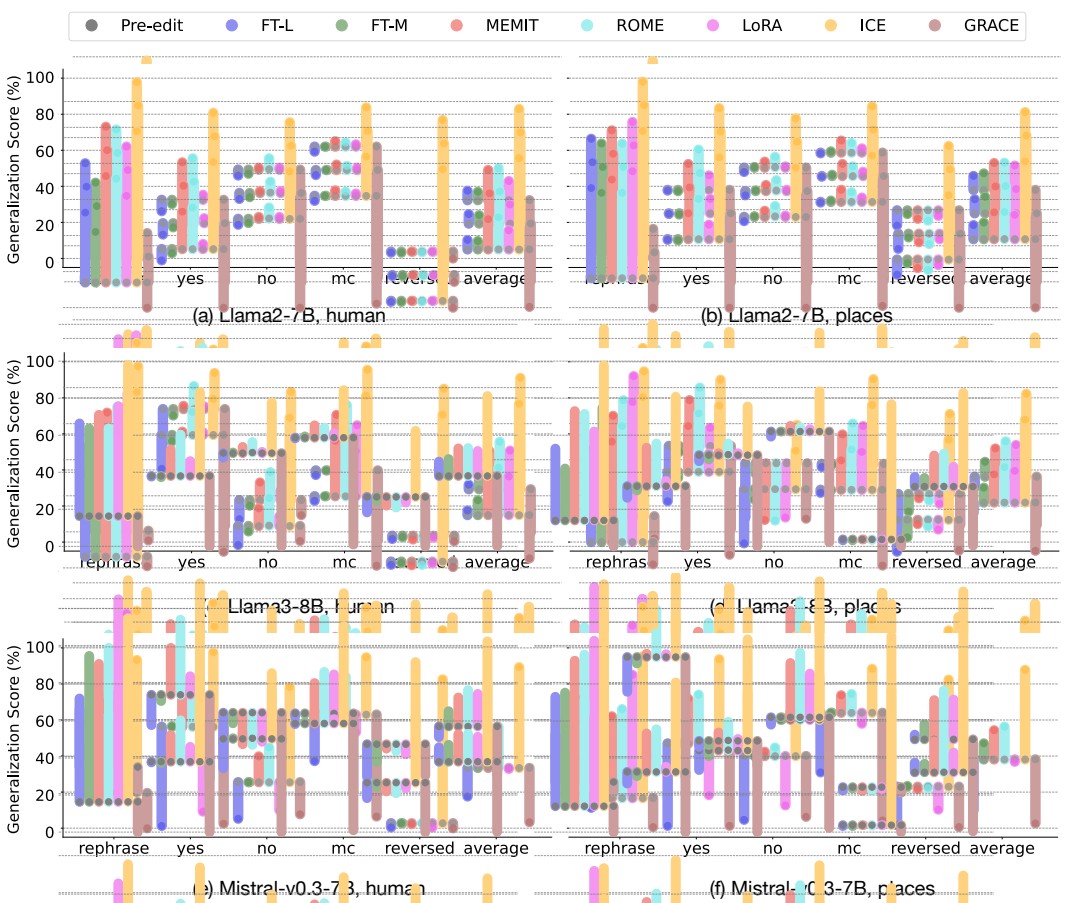

Figure 8: **Generalization Scores of Knowledge Editing Methods on 3 LLMs and 2 Domains**. Generalization Scores (%) are measured by the accuracy on five types of Generalization Evaluation Question-answer Pairs including Rephrased Questions ("rephrase"), two types of Yes-or-No Questions with Yes or No as answers ("yes" or "no"), Multi-Choice Questions ("mc"), Reversed Questions ("reversed"). The "average" refers to the averaged scores over five types of questions. The domains include "human" and "places".

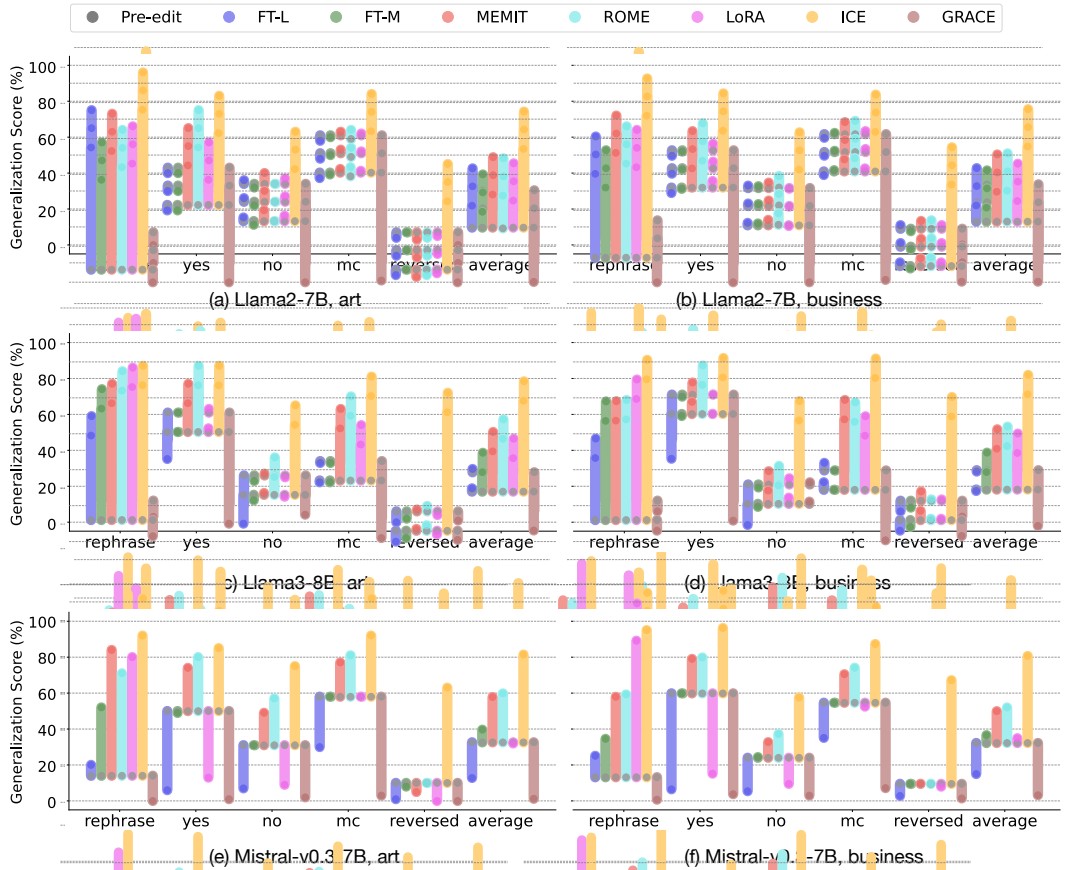

Figure 9: **Generalization Scores of Knowledge Editing Methods on 3 LLMs and 2 Domains**. Generalization Scores (%) are measured by the accuracy on five types of Generalization Evaluation Question-answer Pairs including Rephrased Questions ("rephrase"), two types of Yes-or-No Questions with Yes or No as answers ("yes" or "no"), Multi-Choice Questions ("mc"), Reversed Questions ("reversed"). The "average" refers to the averaged scores over five types of questions. The domains include "art" and "business".

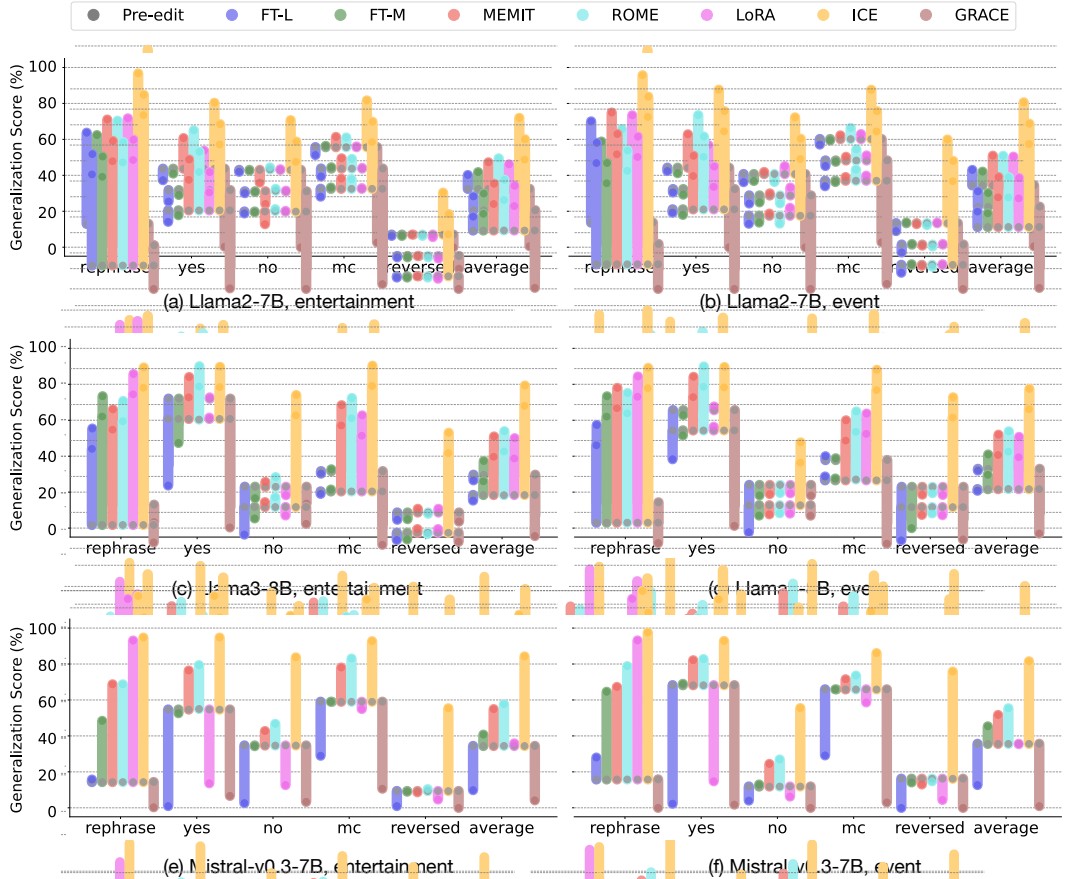

Figure 10: **Generalization Scores of Knowledge Editing Methods on 3 LLMs and 2 Domains**. Generalization Scores (%) are measured by the accuracy on five types of Generalization Evaluation Question-answer Pairs including Rephrased Questions ("rephrase"), two types of Yes-or-No Questions with Yes or No as answers ("yes" or "no"), Multi-Choice Questions ("mc"), Reversed Questions ("reversed"). The "average" refers to the averaged scores over five types of questions. The domains include "entertainment" and "event".

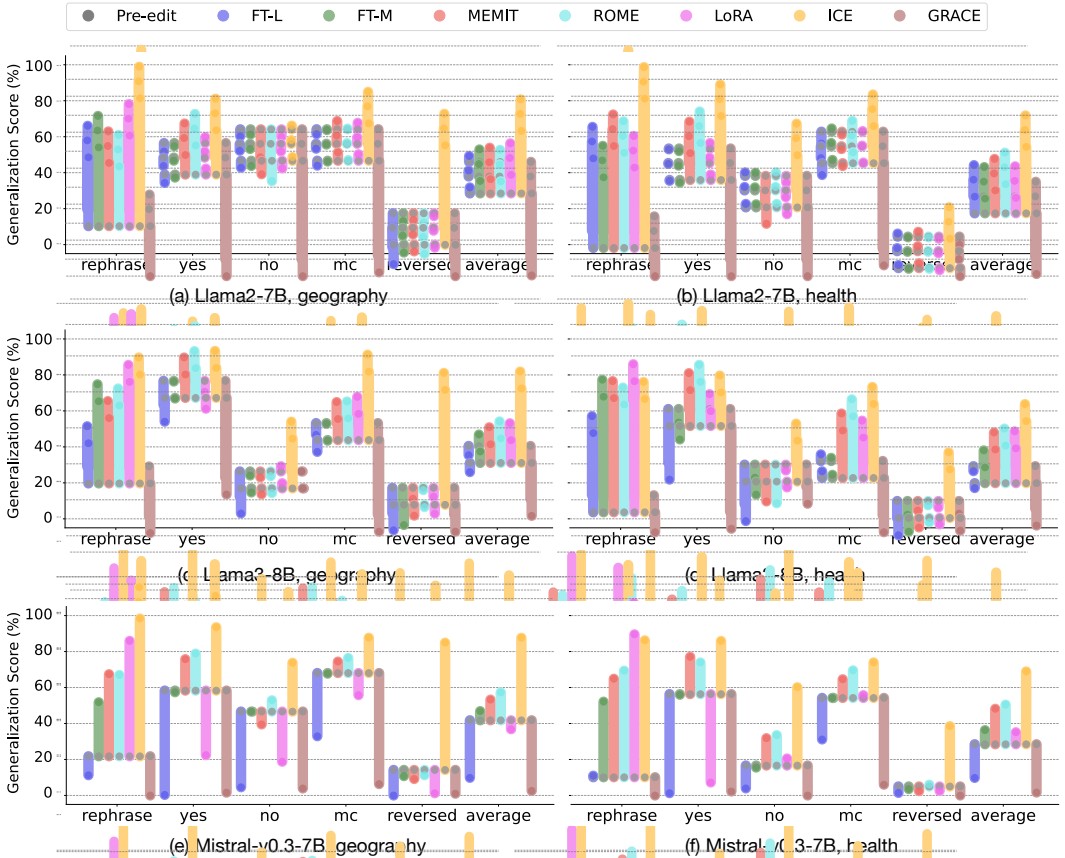

Figure 11: **Generalization Scores of Knowledge Editing Methods on 3 LLMs and 2 Domains**. Generalization Scores (%) are measured by the accuracy on five types of Generalization Evaluation Question-answer Pairs including Rephrased Questions ("rephrase"), two types of Yes-or-No Questions with Yes or No as answers ("yes" or "no"), Multi-Choice Questions ("mc"), Reversed Questions ("reversed"). The "average" refers to the averaged scores over five types of questions. The domains include "geography" and "health".

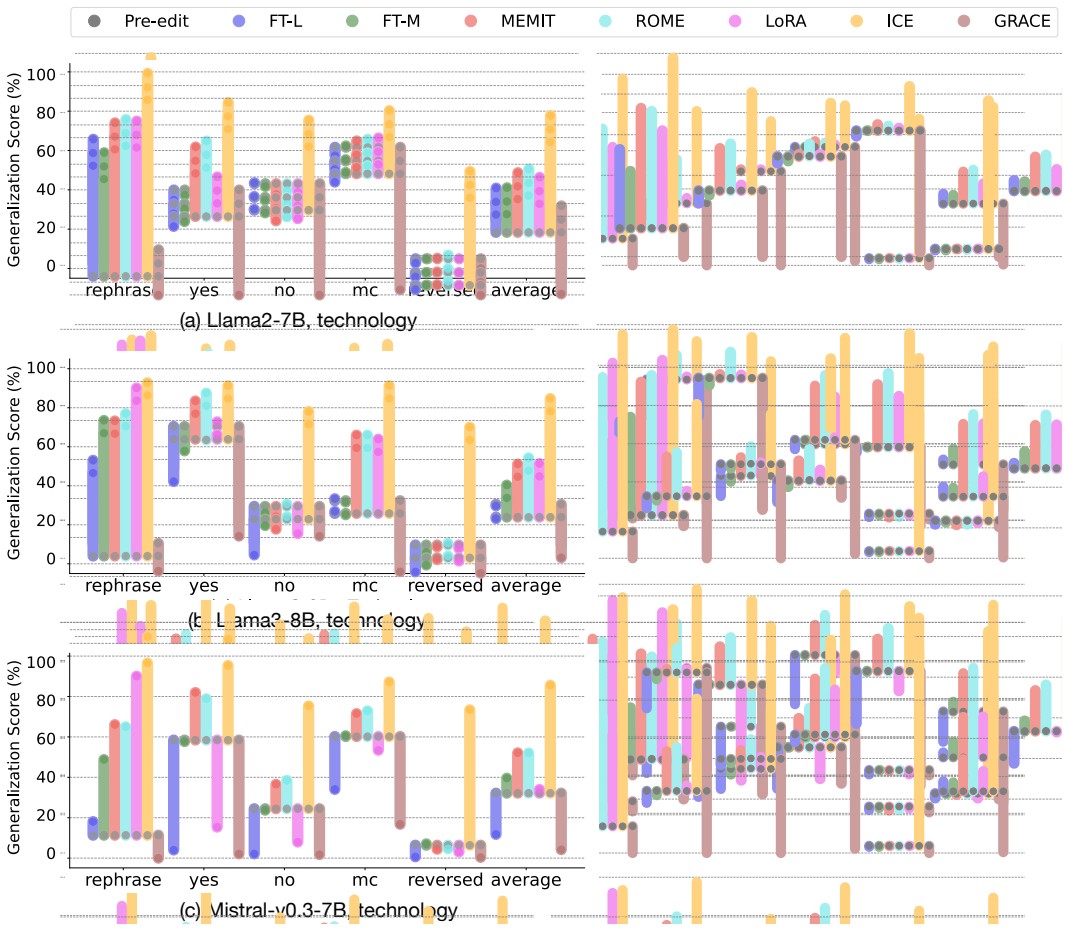

Figure 12: **Generalization Scores of Knowledge Editing Methods on 3 LLMs and 2 Domains**. Generalization Scores (%) are measured by the accuracy on five types of Generalization Evaluation Question-answer Pairs including Rephrased Questions ("rephrase"), two types of Yes-or-No Questions with Yes or No as answers ("yes" or "no"), Multi-Choice Questions ("mc"), Reversed Questions ("reversed"). The "average" refers to the averaged scores over five types of questions. The domain is "technology".

## E.2 Portability Scores of Knowledge Editing Methods on More Domains

Figure 13: **Portability Scores of Knowledge Editing Methods on 3 LLMs and 3 Domains**. Portability Scores (%) are measured by the accuracy on Portability Evaluation Questions, which are Efficacy Evaluation Questions when with $N$ hops. The Portability Evaluation Questions are the same as Efficacy Evaluation Questions when $N$ is 1. The domains include "business", "entertainment", and "event".

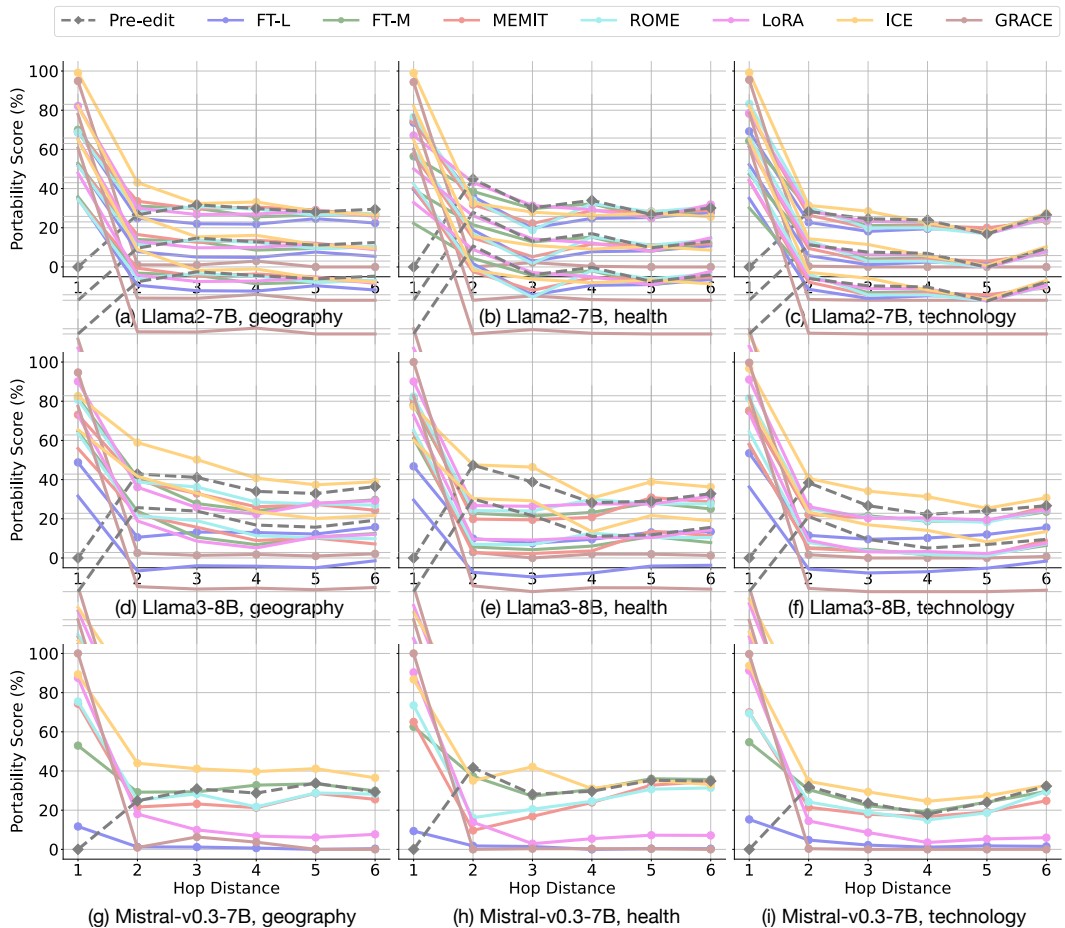

Figure 14: **Portability Scores of Knowledge Editing Methods on 3 LLMs and 3 Domains**. Portability Scores (%) are measured by the accuracy on Portability Evaluation Questions, which are Efficacy Evaluation Questions when with $N$ hops. The Portability Evaluation Questions are the same as Efficacy Evaluation Questions when $N$ is 1. The domains include "geography", "health", and "technology".

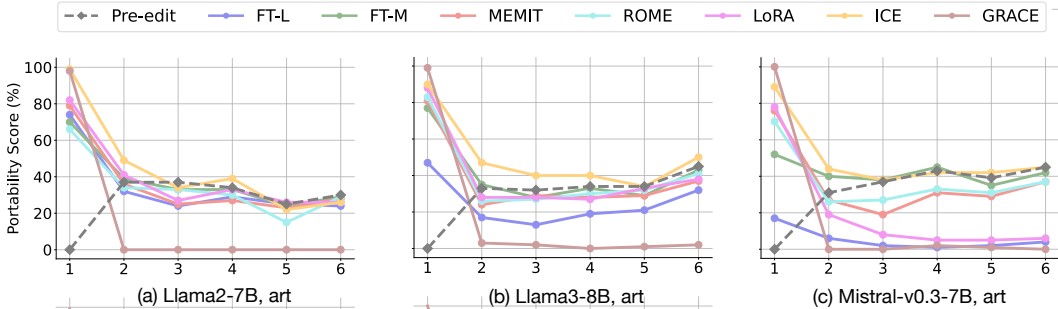

Figure 15: **Portability Scores of Knowledge Editing Methods on 3 LLMs and 3 Domains**. Portability Scores (%) are measured by the accuracy on Portability Evaluation Questions, which are Efficacy Evaluation Questions when with $N$ hops. The Portability Evaluation Questions are the same as Efficacy Evaluation Questions when $N$ is 1. The domain is "art".

### E.3 ROBUSTNESS SCORES OF KNOWLEDGE EDITING METHODS ON MORE DOMAINS

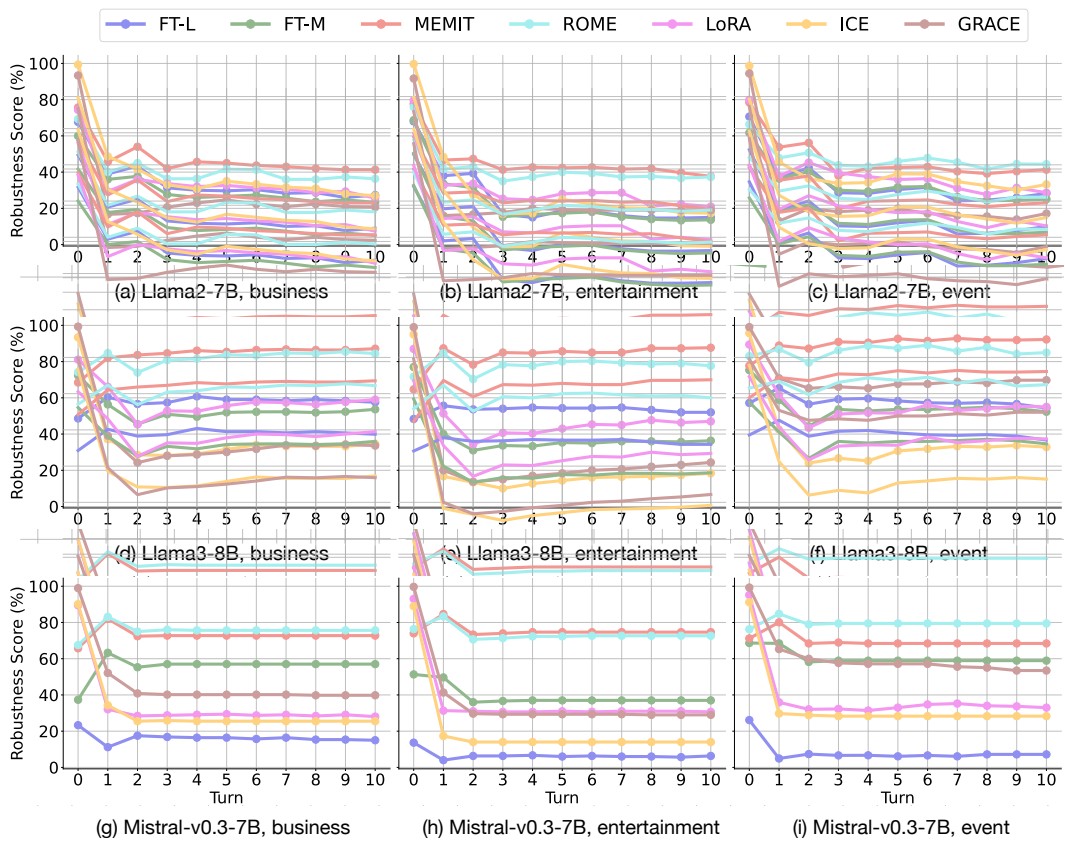

Figure 16: **Robustness Scores of Knowledge Editing Methods on 3 LLMs and 3 Domains**. Robustness Scores are calculated by the accuracy on Robustness Evaluation Questions with $M$ turns ($M = 1 \sim 10$). We regard Efficacy Scores as the Robustness Scores when $M$ is 0. The domains include "business", "entertainment", and "event".

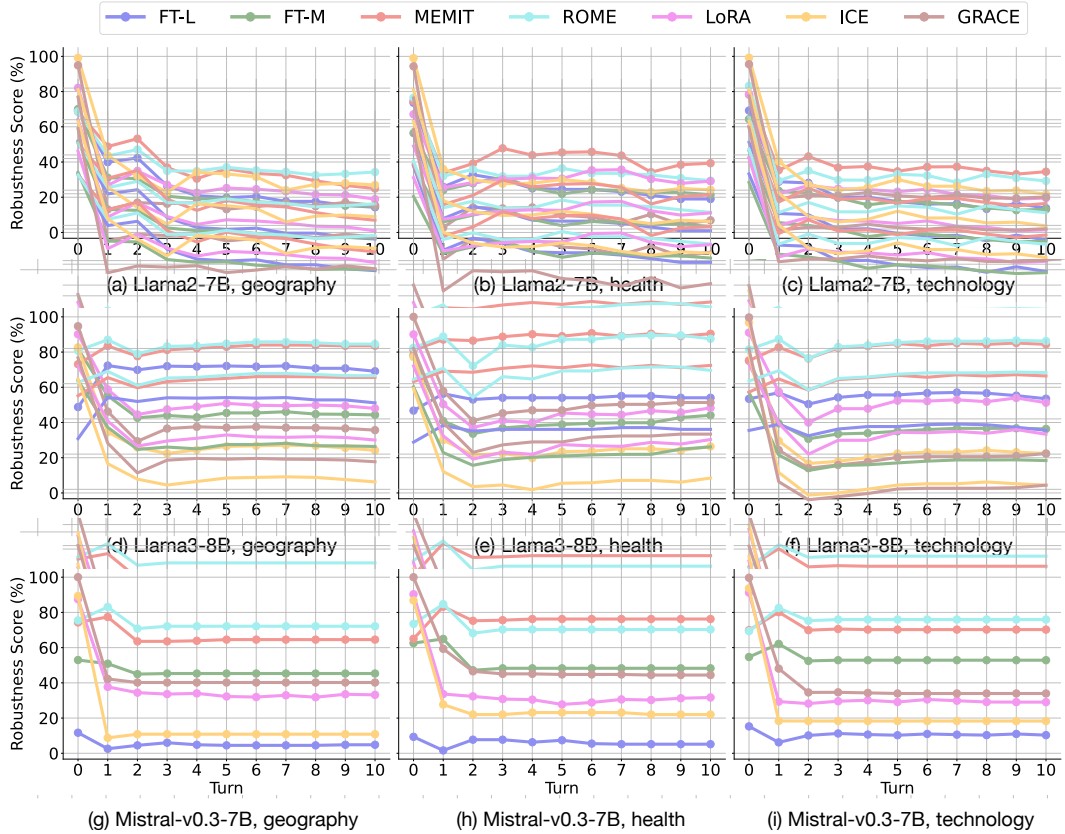

Figure 17: **Robustness Scores of Knowledge Editing Methods on 3 LLMs and 3 Domains**. Robustness Scores are calculated by the accuracy on Robustness Evaluation Questions with $M$ turns ($M = 1 \sim 10$). We regard Efficacy Scores as the Robustness Scores when $M$ is 0. The domains include "geography", "health", and "technology".

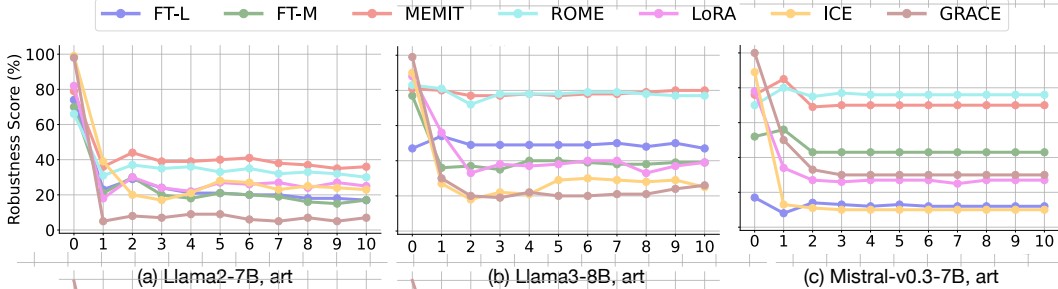

Figure 18: **Robustness Scores of Knowledge Editing Methods on 3 LLMs and 3 Domains**. Robustness Scores are calculated by the accuracy on Robustness Evaluation Questions with $M$ turns ($M = 1 \sim 10$). We regard Efficacy Scores as the Robustness Scores when $M$ is 0. The domain is "art".

# F    EXAMPLES OF HalluEditBench

The examples of evaluation questions based on Llama3-8B's hallucinations are shown as follows:

---

Domain: Places, Topic: Country, Knowledge Triplet: (Trinidad and Tobago, highest point, El Cerro del Aripo)

**Efficacy Evaluation Questions**: What is the highest point of Trinidad and Tobago?
Ground-truth Answer: El Cerro del Aripo
Hallucinated Answer of Llama3-8B before editing: Pierrepont Hill

**Generalization Evaluation Questions**:
Rephrased Question: What is the name of the highest peak in Trinidad and Tobago?
Ground-truth Answer: El Cerro del Aripo

Yes-or-No Questions with Yes as answer: Is El Cerro del Aripo the highest point of Trinidad and Tobago?
Ground-truth Answer: Yes

Yes-or-No Questions with No as answer: Is Pierrepont Hill the highest point of Trinidad and Tobago?
Ground-truth Answer: No

Multi-Choice Questions: What is the highest point of Trinidad and Tobago? A. Mount Kilimanjaro B. Mount Everest C. El Cerro del Aripo D. Pierrepont Hill
Ground-truth Answer: C

Reversed Questions: Which country has El Cerro del Aripo as its highest point?
Ground-truth Answer: Trinidad and Tobago

**Portability Evaluation Questions**:
One-hop question: What is the highest point of Trinidad and Tobago?
Ground-truth Answer: El Cerro del Aripo

Two-hop question: In which mountain range is the highest point of Trinidad and Tobago located?
Ground-truth Answer: Northern Range

Three-hop question: Which island is the mountain range located in that contains the highest point of Trinidad and Tobago?
Ground-truth Answer: Trinidad

Four-hop question: What is the capital city of the island where the mountain range is located that contains the highest point of Trinidad and Tobago?
Ground-truth Answer: Port of Spain

Five-hop question: Which body of water is the capital city near where the mountain range is located that contains the highest point of Trinidad and Tobago?
Ground-truth Answer: Gulf of Paria

Six-hop question: Which ocean is the body of water part of that is near the capital city where the mountain range is located that contains the highest point of Trinidad and Tobago?
Ground-truth Answer: Atlantic Ocean

**Locality Evaluation Questions**: What is the capital city of Trinidad and Tobago?

**Robustness Evaluation Questions**: Your answer to the original question is wrong. Is El Cerro del Aripo the highest point of Trinidad and Tobago? Respond with "Yes" or "No"
Ground-truth Answer: Yes

---

Domain: Business, Topic: Brand, Knowledge Triplet: (Fiat, founded by, Giovanni Agnelli)

**Efficacy Evaluation Questions**: Who was Fiat founded by?
Ground-truth Answer: Giovanni Agnelli
Hallucinated Answer of Llama3-8B before editing: Vincenzo

**Generalization Evaluation Questions**:
Rephrased Question: Who is the founder of Fiat?
Ground-truth Answer: Giovanni Agnelli

Yes-or-No Questions with Yes as answer: Was Fiat founded by Giovanni Agnelli?
Ground-truth Answer: Yes

Yes-or-No Questions with No as answer: Was Fiat founded by Vincenzo?
Ground-truth Answer: No

Multi-Choice Questions: Who was Fiat founded by? A. Giovanni Agnelli B. Enzo Ferrari C. Henry Ford D. Vincenzo
Ground-truth Answer: A

Reversed Questions: Which company was founded by Giovanni Agnelli?
Ground-truth Answer: Fiat

**Portability Evaluation Questions**:
One-hop question: Who was Fiat founded by?
Ground-truth Answer: Giovanni Agnelli

Two-hop question: What is the nationality of the person who founded Fiat?
Ground-truth Answer: Italian

Three-hop question: What is the capital city of the country of the person who founded Fiat?
Ground-truth Answer: Rome

Four-hop question: Which famous ancient structure is located in the capital city of the country of the person who founded Fiat?
Ground-truth Answer: Colosseum

Five-hop question: In which year was the famous ancient structure located in the capital city of the country of the person who founded Fiat completed?
Ground-truth Answer: 80 AD

Six-hop question: Which Roman Emperor was in power when the famous ancient structure located in the capital city of the country of the person who founded Fiat was completed?
Ground-truth Answer: Emperor Titus

**Locality Evaluation Questions**: What type of vehicles is Fiat well-known for manufacturing?

**Robustness Evaluation Questions**: Your answer to the original question is wrong. Was Fiat founded by Giovanni Agnelli? Respond with "Yes" or "No"
Ground-truth Answer: Yes

