# OpenReview forum: "Can Knowledge Editing Really Correct Hallucinations?"
_ICLR.cc/2025/Conference — ICLR 2025 Poster_

### Official Review · Reviewer_jvnV · 2024-10-21

**Soundness:** 3
**Presentation:** 3
**Contribution:** 2
**Rating:** 5
**Confidence:** 4

**Summary:**

This work addresses the prevalent issue of hallucinations in LLMs, where non-factual information is generated despite their impressive performance in various tasks. Knowledge editing has emerged as a solution to correct factual inaccuracies in LLMs without requiring complete retraining. However, current evaluation datasets often do not verify whether LLMs actually produce hallucinated outputs before editing, limiting the ability to assess how well these methods address hallucinations. The authors propose HalluEditBench, a comprehensive benchmark specifically designed to evaluate knowledge editing techniques in correcting real-world hallucinations. The benchmark features a large, diverse dataset spanning 9 domains, 26 topics, and over 6,000 hallucinations, and evaluates methods based on five dimensions: Efficacy, Generalization, Portability, Locality, and Robustness. The paper offers insights into the strengths and weaknesses of current knowledge editing methods, which could drive further advancements in the field.

**Strengths:**

This paper provides a significant contribution by developing a comprehensive and well-structured benchmark tailored to the real-world issue of hallucinations in LLMs. The HalluEditBench framework not only fills a critical gap in current evaluation methodologies but also presents a rigorous, multi-dimensional approach for assessing knowledge editing techniques. Its extensive dataset and focus on practical evaluation criteria (e.g., Efficacy, Generalization) ensure its relevance and applicability in real-world scenarios. This research has the potential to foster future innovations in knowledge editing, addressing a key challenge in improving LLM reliability.

**Weaknesses:**

1. The benchmark proposed in this paper seems to only evaluate specific models, meaning that when models are updated or replaced, the data will need to be reconstructed. If this is indeed the case, it represents the biggest flaw of this work, making the benchmark impractical.
2. The introduction of the five scores lacks formulas explaining how they are calculated, and the purely textual description makes it difficult to understand.
3. The depth of analysis throughout the paper is insufficient, as it only summarizes and briefly discusses the experimental results. I suggest adding a section that analyzes the reasons behind the limitations of existing editing methods (e.g., why Locality Scores of editing methods except FT-M and ICE are unsatisfactory and why ICE and GRACE potentially have low Robustness).
4. When the authors or the publication are included in the sentence, the citation should not be in parenthesis using \citet{} instead of \citep{} (e.g., Line#52, Line#148).
5. Some grammatical errors, e.g., the last sentence in the Abstract should not include 'at', and Line#132 should use 'a knowledge editing operation' instead of 'an knowledge editing operation'. Please carefully review your text to ensure there are no additional grammatical errors.

**Questions:**

1. In Figure 4, why is the Generalization Score of GRACE less than 0 in some cases (e.g., types except 'mc' for Llama2-7B)?
2. ICE and GRACE demonstrate good Efficacy Scores, but their Accuracy is not shown in Table 1. I’m curious about how effective their edits are.
3. Are the five scores in the evaluation all newly proposed in this paper? There are no references to previous evaluation work in Section 2.2.

---

> ### Author Response · Authors · 2024-11-22
>
> We sincerely appreciate your constructive and detailed feedback on our manuscript. Your comments regarding the adaptability of the benchmark, the depth of analysis, and the clarity of evaluation metrics have been instrumental in guiding our revisions. We have worked to address these concerns and provide a more thorough and transparent discussion of our methodology and results. Your observations have motivated us to improve the robustness and presentation of our work, and we are grateful for your thoughtful insights.
>
> \
> *The benchmark proposed in this paper seems to only evaluate specific models, meaning that when models are updated or replaced, the data will need to be reconstructed. If this is indeed the case, it represents the biggest flaw of this work, making the benchmark impractical.*
> - Different models exhibit unique hallucination patterns, making it necessary to tailor datasets for precise evaluation.
> - The main contribution of this paper lies in the insights into various editing methods, which are generalizable across models.
> - We have revised the manuscript to provide additional details on the data construction process in Section 2.1, demonstrating its adaptability. Additional datasets for new models have been added to our GitHub repository (https://anonymous.4open.science/r/hallucination-9D6), and we are committed to releasing datasets for upcoming models.
>
> \
> *The introduction of the five scores lacks formulas explaining how they are calculated, and the purely textual description makes it difficult to understand.
> The depth of analysis throughout the paper is insufficient, as it only summarizes and briefly discusses the experimental results. I suggest adding a section that analyzes the reasons behind the limitations of existing editing methods (e.g., why Locality Scores of editing methods except FT-M and ICE are unsatisfactory and why ICE and GRACE potentially have low Robustness).
> Are the five scores in the evaluation all newly proposed in this paper? There are no references to previous evaluation work in Section 2.2.*
> - The formulas for calculating the five scores are standard accuracy metrics, applied to different subsets of evaluation questions. This clarification has been added to Section 2.2.
> - These scores are commonly used in knowledge editing evaluation, as referenced in prior works (e.g., [1], [2], [3], [4]).
> - We acknowledge the depth of analysis could be expanded. While our focus is on summarizing general trends (e.g., differences between parameter-preserving and parameter-modifying methods), we plan to provide more detailed analysis in future work.
>
> \
> *When the authors or the publication are included in the sentence, the citation should not be in parenthesis using \citet{} instead of \citep{} (e.g., Line#52, Line#148). Some grammatical errors, e.g., the last sentence in the Abstract should not include 'at', and Line#132 should use 'a knowledge editing operation' instead of 'an knowledge editing operation'. Please carefully review your text to ensure there are no additional grammatical errors.*
> - We have corrected the citation format and addressed the typos.
>
> \
> *In Figure 4, why is the Generalization Score of GRACE less than 0 in some cases (e.g., types except 'mc' for Llama2-7B)?*
> - The original visualization caused a misinterpretation (the round edges slightly pass the bottom line due to the size of the circle). We have updated Figure 4, and Figure 8 to 12. for clarity. None of the values are below zero; the issue arose from graphical overlap.
>
> \
> *ICE and GRACE demonstrate good Efficacy Scores, but their Accuracy is not shown in Table 1. I’m curious about how effective their edits are.*
> - The reason we didn’t include results for GRACE is that the paper we are comparing to (Table 4 in [1]) didn’t include GRACE in their evaluation. As for ICE, we use a more simple variant of IKE used in [1]. ICE use vanilla in context learning with no demonstrations and embedding cache as in IKE. The results and code for IKE is include in our repo, below is a table including IKE (code/result_table_previous_benchmark.ipynb)
> | Method            | WikiData_recent  |  ZsRE    | WikiBio |
> | :---------------- | :----------------------------------: | :-----: | :-----: |
> | Pre-edit          | 47\.40                               | 37\.49  | 61\.35  |
> | Post-edit (ROME)  | 97\.37                               | 96\.86  | 95\.91  |
> | Post-edit (MEMIT) | 97\.10                               | 95\.86  | 94\.68  |
> | Post-edit (FT-L)  | 56\.30                               | 53\.82  | 66\.70  |
> | Post-edit (FT-M)  | 100\.00                              | 99\.98  | 100\.00 |
> | Post-edit (LoRA)  | 100\.00                              | 100\.00 | 100\.00 |
> | Post-edit (IKE)  | 99\.97                              | 99\.84 | 100\.00 |

---

> > ### Author Response · Authors · 2024-11-22
> >
> > *The paper’s experiments focus on a few smaller LLMs (e.g., Llama3-8b, Mistral-v0.2-7b), limiting the findings' applicability to larger, state-of-the-art models that may respond differently to editing attacks. This narrow scope weakens the generalizability and robustness of the conclusions. For instance, ICE experiments could be conducted on more advanced models, such as Gemini or GPT-4o.*
> > - We focused on open-weight models for reproducibility and comparability. Experiments on proprietary models like Gemini or GPT-4o are challenging due to their limited accessibility and reproducibility.
> > - Evaluation on larger models and proprietary models is difficult due to the limitation of computing resources and API credits. Given that recent studies [5, 6] still use older models such as T5, GPT-2, and GPT-J, our choice of LLMs is relatively up-to-date.
> > - Our approach can be easily adapted to newer models, and we are committed to expanding our dataset to include results for state-of-the-art open-weight models.
> >
> > \
> > [1] Zhang, Ningyu, Yunzhi Yao, Bozhong Tian, Peng Wang, Shumin Deng, Mengru Wang, Zekun Xi et al. "A comprehensive study of knowledge editing for large language models." arXiv preprint arXiv:2401.01286 (2024).\
> > [2] Xie, Qiming, Zengzhi Wang, Yi Feng, and Rui Xia. "Ask Again, Then Fail: Large Language Models' Vacillations in Judgement." arXiv preprint arXiv:2310.02174 (2023).\
> > [3] Xu, Rongwu, Brian S. Lin, Shujian Yang, Tianqi Zhang, Weiyan Shi, Tianwei Zhang, Zhixuan Fang, Wei Xu, and Han Qiu. "The Earth is Flat because...: Investigating LLMs' Belief towards Misinformation via Persuasive Conversation." arXiv preprint arXiv:2312.09085 (2023).\
> > [4] Wang, Song, Yaochen Zhu, Haochen Liu, Zaiyi Zheng, Chen Chen, and Jundong Li. "Knowledge editing for large language models: A survey." ACM Computing Surveys (2023).\
> > [5] Tan, Chenmien, Ge Zhang, and Jie Fu. "Massive editing for large language models via meta learning." ICLR (2024).\
> > [6] Hartvigsen, Tom, Swami Sankaranarayanan, Hamid Palangi, Yoon Kim, and Marzyeh Ghassemi. "Aging with grace: Lifelong model editing with discrete key-value adaptors." Advances in Neural Information Processing Systems 36 (2024).

---

> > ### Comment · Reviewer_jvnV · 2024-11-22
> > **Official Comment by Reviewer jvnV**
> >
> > Thank you for the detailed clarifications, which have addressed some of my concerns, and I have raised my score. However, I remain concerned about the issue mentioned in Weakness 3 (which I noticed was also pointed out by Reviewer UgrK). Some of the analysis should be conducted within this paper rather than deferred to future work. If the depth of analysis can be expanded, I feel the paper will be on the borderline.

---

> ### Author Response · Authors · 2024-11-22
> **Thanks for the quick reply! We sincerely appreciate your suggestions, and would like to clarify that we actually not only present the observations but also provide analysis and explanations in the paper.**
>
> Thanks for the quick reply! We sincerely appreciate your suggestions, and would like to clarify that **we actually not only present the observations but also provide analysis and explanations in the paper**.
> - As for **Efficacy**, we notice ICE and GRACE clearly outperform the other editing methods, which all need to modify the original parameters of LLMs. *The potential explanation is that parameter-preserving editing methods have intrinsic advantage over parameter-modifying editing methods.*
> - As for **Generalization**, we observe that the pre-edit Generalization Scores are not 0 even though their pre-edit Efficacy Scores are ensured 0. *The potential reason is that the manifestation of hallucination actually depends on the design of question prompts.*
> - As for **Portability**, we observe that the pre-edit Portability Scores are not zero for 2 ∼ 6 hops even though the Portability Score is 0 for 1 hop. *The potential reason is that LLMs may memorize answers rather than reason based on single-hop knowledge for multi-hop questions, which further causes that LLMs may not really reason with edited knowledge in multi-hop questions.*
> - As for **Locality**, we notice that there are high fluctuations across domains and LLMs. *The potential reason is that domains and LLMs have a high impact on the Locality Scores of knowledge editing methods.*
> - As for **Robustness**, we observe that both ICE and GRACE have a low level of robustness. *The hypothesis is that parameter-preserving editing methods are intrinsically susceptible to distractions in the context.*
>
> We also would like to emphasize our contribution of **a new massive evaluation dataset for the field of knowledge editing**, which has addressed the pressing need of using real-world LLM hallucinations to assess knowledge editing techniques, and **the  three key findings**: 1. The effectiveness of knowledge editing methods in correcting real-world hallucinations could be far from what their performance on existing datasets suggests. 2. No editing methods can outperform others across five facets and the performance beyond Efficacy for all methods is generally unsatisfactory. 3. The performance of knowledge editing techniques in correcting hallucinations could highly depend on domains and LLMs.
>
> **We acknowledge that more experiments and studies are desired to fully explain all the new findings in our paper, which may be beyond the scope of our paper.** We believe our observations as well as the preliminary analysis and explanations can facilitate the progress of the field of knowledge editing and inspire more future improvements.

---

> ### Author Response · Authors · 2024-12-03
> **Dear Reviewer, if you have already found our responses satisfactory, we humbly remind you of a fitting update of the final rating. Thanks for your time and effort again!**
>
> Dear Reviewer jvnV,
>
> We are genuinely grateful for your constructive feedback and the acknowledgement of our contributions. if you have already found our responses satisfactory, we humbly remind you of a fitting update of the final rating. Thanks for your time and effort again!
>
> The authors

---

> > ### Comment · Reviewer_jvnV · 2024-12-03
> > **Official Comment by Reviewer jvnV**
> >
> > Thanks for your response. I think my rating is final.

---

> > > ### Author Response · Authors · 2024-12-03
> > > **Could you let us know your remaining concern? We are happy to provide more explanations. Thanks!**
> > >
> > > Dear Reviewer jvnV,
> > >
> > > Thanks for the quick reply! We hope your concern about Weakness 3 has been addressed. Could you let us know your remaining concern? We are happy to provide more explanations. Thanks!
> > >
> > > The authors

---

### Official Review · Reviewer_UgrK · 2024-11-03

**Soundness:** 3
**Presentation:** 3
**Contribution:** 2
**Rating:** 6
**Confidence:** 3

**Summary:**

While knowledge editing aims to correct these inaccuracies without full retraining, existing evaluation datasets often do not verify if LLMs generate hallucinated responses before editing. The authors present HalluEditBench, a benchmarking framework that evaluates knowledge editing methods against a dataset of over 6,000 hallucinations across 9 domains and 26 topics. Performance is assessed on five dimensions: Efficacy, Generalization, Portability, Locality, and Robustness. The findings offer insights into the strengths and limitations of various editing techniques, contributing to advancements in the field.

**Strengths:**

S1. This paper presents HalluEditBench, a benchmarking framework that evaluates knowledge editing methods against a dataset of over 6,000 hallucinations across 9 domains and 26 topics. This facilitates future research on leveraging knowledge editing techniques to mitigate hallucinations in LLMs.

S2. This paper introduces some novel insights, such as: The current assessment of knowledge editing could be unreliable; The manifestation of hallucination depends on question design; Editing methods marginally improve or degrade pre-edit Portability Scores, implying LLMs may not really reason with edited knowledge in multi-hop questions.; Efficacy does not have a noticeable correlation with Locality; Parameter-preserving knowledge editing methods such as ICE and GRACE potentially have low Robustness.

S3. The paper is well-organized and has good readability.

**Weaknesses:**

W1. The paper mentions that existing datasets for knowledge editing fail to verify whether knowledge editing methods can effectively correct hallucinations in large language models. The paper should provide at least one example to illustrate the shortcomings of other benchmarks in this regard. For instance, it could include statistics from other datasets or highlight differences in dataset construction methods compared to the approach proposed here, underscoring the innovations introduced by this work in hallucination detection. This would help clarify the paper’s contributions.

W2. Secondly, when presenting certain insights, the paper should include some speculation on the causes of these insights to enhance their plausibility and persuasiveness. For example, in Section 3.1, the differing performance of the FT-M method on two distinct datasets, and in Section 3.2, the significant drop in the GRACE method's performance on the generalization metric

**Questions:**

The paper should place greater emphasis on why existing benchmarks fail to validate the effectiveness of knowledge editing methods in addressing hallucinations. Additionally, it should provide a deeper analysis of the insights presented.

---

> ### Author Response · Authors · 2024-11-22
>
> Thank you for your thorough review and for providing constructive suggestions that have enriched our understanding of how to better communicate our contributions. Your emphasis on clarifying the shortcomings of previous benchmarks and expanding our analysis of insights has been invaluable. We have made significant efforts to incorporate your feedback into the revised manuscript and believe these changes provide a more comprehensive and persuasive presentation of our findings. Your thoughtful feedback has been a great help in advancing the quality of this work.
>
> \
> *W1. The paper mentions that existing datasets for knowledge editing fail to verify whether knowledge editing methods can effectively correct hallucinations in large language models. The paper should provide at least one example to illustrate the shortcomings of other benchmarks in this regard. For instance, it could include statistics from other datasets or highlight differences in dataset construction methods compared to the approach proposed here, underscoring the innovations introduced by this work in hallucination detection. This would help clarify the paper’s contributions.*
> - There are differences in statistics, question types, and data sources between prior datasets and the proposed ones. However, the key distinction lies in tailoring datasets specifically for each LLM, as detailed in Table below.
> - Unlike prior datasets (e.g., zsRE, CounterFact), which do not ensure pre-edit accuracy of zero, our pipeline generates responses for all collected factual questions and only keeps those that the target LLM cannot answer correctly. This ensures a more reliable evaluation of knowledge editing methods.
>
> \
> *W2. Secondly, when presenting certain insights, the paper should include some speculation on the causes of these insights to enhance their plausibility and persuasiveness. For example, in Section 3.1, the differing performance of the FT-M method on two distinct datasets, and in Section 3.2, the significant drop in the GRACE method's performance on the generalization metric*
>
> We sincerely appreciate your suggestions, and would like to clarify that **we actually not only present the observations but also provide analysis and explanations in the paper**.
> - As for **Efficacy**, we notice ICE and GRACE clearly outperform the other editing methods, which all need to modify the original parameters of LLMs. *The potential explanation is that parameter-preserving editing methods have intrinsic advantage over parameter-modifying editing methods.*
> - As for **Generalization**, we observe that the pre-edit Generalization Scores are not 0 even though their pre-edit Efficacy Scores are ensured 0. *The potential reason is that the manifestation of hallucination actually depends on the design of question prompts.*
> - As for **Portability**, we observe that the pre-edit Portability Scores are not zero for 2 ∼ 6 hops even though the Portability Score is 0 for 1 hop. *The potential reason is that LLMs may memorize answers rather than reason based on single-hop knowledge for multi-hop questions, which further causes that LLMs may not really reason with edited knowledge in multi-hop questions.*
> - As for **Locality**, we notice that there are high fluctuations across domains and LLMs. *The potential reason is that domains and LLMs have a high impact on the Locality Scores of knowledge editing methods.*
> - As for **Robustness**, we observe that both ICE and GRACE have a low level of robustness. *The hypothesis is that parameter-preserving editing methods are intrinsically susceptible to distractions in the context.*
>
> We also would like to emphasize our contribution of **a new massive evaluation dataset for the field of knowledge editing**, which has addressed the pressing need of using real-world LLM hallucinations to assess knowledge editing techniques, and **the  three key findings**: 1. The effectiveness of knowledge editing methods in correcting real-world hallucinations could be far from what their performance on existing datasets suggests. 2. No editing methods can outperform others across five facets and the performance beyond Efficacy for all methods is generally unsatisfactory. 3. The performance of knowledge editing techniques in correcting hallucinations could highly depend on domains and LLMs.
>
> **We acknowledge that more experiments and studies are desired to fully explain all the new findings in our paper, which may be beyond the scope of our paper.** We believe our observations as well as the preliminary analysis and explanations can facilitate the progress of the field of knowledge editing and inspire more future improvements.

---

> ### Author Response · Authors · 2024-11-22
>
> *The paper should place greater emphasis on why existing benchmarks fail to validate the effectiveness of knowledge editing methods in addressing hallucinations. Additionally, it should provide a deeper analysis of the insights presented.*
> - Below is a comparison table highlighting the statistical differences between prior benchmarks and our dataset. However, as shown in Table below, the key innovation lies in tailoring the dataset for each model, ensuring pre-edit accuracy of zero.
> - Our data construction pipeline can be easily adapted to new LLMs, as demonstrated by the datasets for GPT-J and Gemma added to our GitHub repository (https://anonymous.4open.science/r/hallucination-9D6).
>
> | Datasets     | Real-world LLM Hallucinations | Size   | Question Type                                                                                                                                      |
> | :------------- | :-----  | :----- | :------------------------------------------------------------------------------------------------------------------------------------------------- |
> | Wikirecent    | No (pre-edit accuracy is not 0) | 1,836  | Efficacy, Portability, Locality                                                                                                                    |
> | ZsRE           | No (pre-edit accuracy is not 0) | 11,301 | Efficacy, Generalization (paraphrase), Portability, Locality                                                                                       |
> | WikiBio        | No (pre-edit accuracy is not 0)  | 1,984  | Efficacy, Locality                                                                                                                                 |
> | HalluEditBench | Yes (pre-edit accuracy is 0)  | 6,738  | Efficacy, Generalization (Paraphrased, Multiple choices, Yes question, No question, Reversed relation question), Portability, Locality, Robustness |

---

> > ### Comment · Reviewer_UgrK · 2024-11-25
> >
> > Your response largely addressed my concerns; thank you for your reply.
> > Please include the additional tables in the appendix of future versions.
> > Some inconclusive experimental observations can be added to the limitations section to guide future researchers.

---

> > > ### Author Response · Authors · 2024-11-25
> > > **Thanks for your constructive feedback and the acknowledgement of our contributions!**
> > >
> > > Dear Reviewer UgrK,
> > >
> > > We are genuinely grateful for your constructive feedback and the acknowledgement of our contributions. We will follow your suggestions in the revision. if you have already found our responses satisfactory, we humbly remind you of a fitting update of the final rating. Thanks for your time and effort again!
> > >
> > > The authors

---

### Official Review · Reviewer_Jeec · 2024-11-03

**Soundness:** 4
**Presentation:** 3
**Contribution:** 3
**Rating:** 8
**Confidence:** 3

**Summary:**

As LLMs are now in productive use for a variety of tasks, the demand for the reliability of their output is growing. However, it is well documented that language models do not always provide reliable facts as output, partly because they usually remain at the level of knowledge of their training and some facts change over time, and partly because it is known that language models can also provide output that is not fact-based. These so called hallucinations are a major problem and various suggestions have already been made as to how models can be prevented from hallucinating.
Knowledge editing methods, for example, are used to update the existing knowledge of a model. This involves updating facts in the form of knowledge triplets by exchanging the object. These methods can also be used for corrections and have already been successfully used to correct the knowledge of models with the corresponding data sets / knowledge bases.
However, as the authors of the article correctly point out, these corrections do not take into account what the models were hallucinating about before they were corrected, but can only determine a general improvement.
Therefore, they create a data set of hallucinations from three models and compare the results of current knowledge editing techniques for five different aspects.
The results of the experiments are as interesting as they are sobering: None of the techniques delivers consistently good results for all models and under all aspects, but the divergent results lead to interesting insights.

**Strengths:**

A major strength of the paper is clearly that the results presented show  that the effectiveness of knowledge editing techniques can vary greatly and is not necessarily as great on actual hallucinations of the LLMs as the experiments on the previous existing data sets show. The fact that the domain of the fact and the model itself have a decisive impact on the  efficiency (and other scores) is an important point and should be taken into account. The article convincingly and clearly demonstrates that none of the techniques tested can claim to be a satisfactory solution for updating knowledge from diverse domains and from all types of language models and thus for preventing hallucinations. The results are clearly presented and well analyzed throughout. The authors' approach and their experiments are comprehensible and explainable.

**Weaknesses:**

I missed the authors answering the question from the title of the article more conclusively or at least clearly addressing it again in their conclusion, as this question can basically be answered in the negative (at least in part) based on the results presented. There is also no corresponding outlook. The summary is therefore somewhat abbreviated.
Although the structure of the paper is good and clearly laid out, the section on related work seems like an appendix. It could be moved to before the second section, especially as it cites the knowledge editing techniques in particular, which are then described and classified in more detail in section 2. I would also suggest moving Figure 2 to the Appendix.
Although the statistics of the data set are of course useful information, it is not so important for the understanding of the article itself whether  the data for Llama2-7B contains 26 hallucinations on the topic "health - medication" vs. 27 on the topic "health - symptom". The figure is also somewhat small for this wealth of information.
The question-answer pairs based on the hallucinations were generated  with GPT-4o, the prompt is documented in the appendix. Although only tasks such as paraphrasing were required, GPT-4o could in the worst case hallucinate itself or deliver incorrect results. Unfortunately, it is not clear and discussed whether this has been ruled out in case I have not overlooked this.
minor remarks:
•	Typo “overfiting”, page 4, section 2.3
•	MIT license file not proper anonymized

**Questions:**

•	Were the question-answer pairs generated by GPT-4o checked manually for correctness?
•	Are the benchmark results still comparable if it may not be possible to draw a similar distribution of hallucinations across the various subject areas for a specific model?

---

> ### Author Response · Authors · 2024-11-22
>
> We greatly appreciate your insightful feedback and the time you have taken to evaluate our work. We have carefully addressed these concerns and made revisions to ensure the manuscript. Thank you for highlighting these important aspects, which have significantly helped us enhance the quality and readability of our paper.
>
> \
> *I missed the authors answering the question from the title of the article more conclusively or at least clearly addressing it again in their conclusion, as this question can basically be answered in the negative (at least in part) based on the results presented. There is also no corresponding outlook. The summary is therefore somewhat abbreviated.*
> - The title question requires a case-by-case analysis based on the specific editing methods, models, and topics involved. We strive to extract common findings across different methods, as detailed in Section 3.
> - To address this feedback, we have expanded the conclusion to explicitly summarize the key insights from the study and offer a clearer outlook for future research directions.
>
> \
> *Although the structure of the paper is good and clearly laid out, the section on related work seems like an appendix. It could be moved to before the second section, especially as it cites the knowledge editing techniques in particular, which are then described and classified in more detail in section 2. I would also suggest moving Figure 2 to the Appendix. Although the statistics of the data set are of course useful information, it is not so important for the understanding of the article itself whether the data for Llama2-7B contains 26 hallucinations on the topic "health - medication" vs. 27 on the topic "health - symptom". The figure is also somewhat small for this wealth of information.*
> - We have increased the font size and clarity of Figure 2. However, we put it in Section 2 to highlight the comprehensive coverage of our proposed datasets.
> - Regarding the placement of the related work section, we aim to prioritize the discussion of our data construction and findings. The detailed related work is provided in Section 4 and Appendices B and D.
>
> \
> *The question-answer pairs based on the hallucinations were generated with GPT-4o, the prompt is documented in the appendix. Although only tasks such as paraphrasing were required, GPT-4o could in the worst case hallucinate itself or deliver incorrect results. Unfortunately, it is not clear and discussed whether this has been ruled out in case I have not overlooked this.*
> *Were the question-answer pairs generated by GPT-4o checked manually for correctness?*
> - Among all the question types—including Efficacy, Generalization (Paraphrased, Multiple Choice, Yes/No questions, Reversed Relation questions), Portability, Locality, and Robustness—only Portability answers are generated using GPT-4o, while the other answers are derived from factual triplets. We acknowledge the limitation of using GPT-4o for generating multi-hop questions. However, we observed that this approach produces questions with higher logical quality compared to rule-based methods.
> - Previous studies, such as [1, 2], have also highlighted the limitations of rule-based approaches in generating high-quality multi-hop questions.
> [1] Zhong, Zexuan, Zhengxuan Wu, Christopher D. Manning, Christopher Potts, and Danqi Chen. "MQuAKE: Assessing Knowledge Editing in Language Models via Multi-Hop Questions." In Proceedings of the 2023 Conference on Empirical Methods in Natural Language Processing, pp. 15686-15702. 2023.
> [2] MQuAKE-Remastered: Multi-Hop Knowledge Editing Can Only Be Advanced With Reliable Evaluations https://openreview.net/pdf/132633a5c62ea97bf3f7d8ab65b2e8b3cc4947a6.pdf
>
> \
> *minor remarks: • Typo “overfiting”, page 4, section 2.3 • MIT license file not proper anonymized*
> - We have corrected the typo and revised the license file.
>
> \
> *Are the benchmark results still comparable if it may not be possible to draw a similar distribution of hallucinations across the various subject areas for a specific model?*
> - We included a variety of topics to ensure comprehensive coverage, enabling general insights to be drawn. While we speculate that similar results would be comparable even with a different topic distribution, we recognize this as an open question and plan to explore it in future work.

---

### Official Review · Reviewer_Umf8 · 2024-11-04

**Soundness:** 2
**Presentation:** 2
**Contribution:** 3
**Rating:** 5
**Confidence:** 5

**Summary:**

The paper focuses on the assessment of knowledge editing methods in correcting real-world LLM hallucinations. It introduces a new benchmark, HalluEditBench, which evaluates the effectiveness of these techniques specifically for addressing real-world hallucinations. HalluEditBench encompasses a large dataset of verified hallucinations across 9 domains and 26 topics, amounting to over 6,000 instances. The benchmark assesses knowledge editing methods across five dimensions: Efficacy, Generalization, Portability, Locality, and Robustness. Through this framework, the study provides insights into the limitations and strengths of seven established knowledge editing techniques across multiple LLM architectures, guiding further advances in knowledge editing methodologies.

**Strengths:**

The paper offers a foundational contribution to the study of hallucination correction in LLMs, posing a critical question that has been overlooked in the field: Can knowledge editing effectively address LLM hallucinations?

The authors’ approach demonstrates originality by establishing five distinct evaluation facets (Efficacy, Generalization, Portability, Locality, and Robustness), each of which extends beyond conventional measures to comprehensively assess knowledge editing impacts. These dimensions represent a significant innovation that could be widely adopted in broader LLM evaluations, potentially becoming standard practice for assessing post-training improvements in LLMs.

The experimental insights are both intriguing and impactful, with potential to influence not only the field of knowledge editing but also the broader post-training processes in LLM development.

The work presents a substantial and well-designed set of experiments that effectively support and validate the proposed evaluation metrics.

**Weaknesses:**

The paper could benefit from providing a more comprehensive description of the dataset construction process, which currently lacks sufficient detail. There is limited information on how domains and topics were sampled or selected from the knowledge source. The paper does not indicate whether the dataset predominantly features popular entities or long-tail entities (e.g., [Sun et al., 2023](https://arxiv.org/abs/2308.10168)), which may impact the generalizability of findings across different distributions of knowledge. I would suggest the authors provide information on the criteria used for selecting domains and topics and an analysis of the distribution of entity popularity in the dataset.

Since the dataset is built from Wikipedia/Wikidata, which are frequently updated, there is potential for temporal mismatches between the dataset and the LLMs. For instance, some LLMs may have been trained on earlier versions, while others may align with more recent data. The authors might need to quantify and analyze the potential extent of outdated or inconsistent information across different LLMs to strengthen the paper’s validity, or propose a method to estimate the temporal alignment between the dataset and each LLM's training data.

As mentioned in the paper, the hallucinations can happen differently for (1) different models, (2) pre-trained/fine-tuned in different periods, (3) different prompt input, etc. The generalization ability of the benchmark may be limited. Since LLMs' knowledge evolves with new releases, a focus on the dataset construction pipeline could add long-term value beyond the static dataset. The authors should emphasize the utility of their dataset construction pipeline, which might hold greater long-term value than the static dataset itself. I would suggest the authors to provide a detailed description of their dataset construction pipeline, including any code or tools used, to enable reproducibility and extension of their work.

**Questions:**

In the dataset construction section, it is mentioned that Wikipedia was selected as the factual knowledge source, yet the triples were said to be retrieved using the Wikidata Query Service. Given that Wikipedia and Wikidata are distinct resources with content gaps, could the authors clarify where the dataset primarily comes from? Also, since Wikidata is actively edited, please report the version of the dump used for reproducibility.

For completeness, when citing FT-L, it would be helpful to include both Meng et al., 2022, and the original work by Zhu et al., 2020.

Given that 5 out of the 7 knowledge editing techniques evaluated have reported results on GPT family models in their original studies, including at least one GPT model in this study would provide a more comprehensive baseline and strengthen the comparability of results.

---

> ### Author Response · Authors · 2024-11-22
>
> We sincerely thank you for your constructive feedback. Your detailed comments have provided valuable insights that help us to enhance the clarity and rigor of our work, particularly regarding the dataset construction process and the importance of analyzing the implications of topic distribution and temporal mismatches. We have taken your suggestions seriously and have revised the manuscript to address these points. We appreciate your advice to help us refine our work.
>
> \
> *The paper could benefit from providing a more comprehensive description of the dataset construction process, which currently lacks sufficient detail. There is limited information on how domains and topics were sampled or selected from the knowledge source. The paper does not indicate whether the dataset predominantly features popular entities or long-tail entities (e.g., Sun et al., 2023), which may impact the generalizability of findings across different distributions of knowledge. I would suggest the authors provide information on the criteria used for selecting domains and topics and an analysis of the distribution of entity popularity in the dataset.*
> - We have updated the manuscript to include more description of the topic selection process in Section 2.1.
> - Our data construction mainly aims to ensure comprehensiveness and diversity across topics. Domains and topics were chosen based on their availability and the number of triplets in Wikidata. Using SPARQL, we identified various common topics and included only those with more than 100 triplets. This threshold was determined to ensure sufficient data samples for further processing (e.g., we filtered out the triplets that share the same subject and relation while the objects are different, indicating there are more than one answer to questions about the object). We exclude following topics because triplets retrieved using SPARQL are fewer than 100: "invention", "animal species", "mineral", "Olympic Games", "train", "mathematics", "neuroscience", "robotics", "internet", "mobile phone", "3D printing", "bird", "academy awards", "movies", "movie", "grammy award", 'netflix series', 'beverage', "climate", "astronomy", "climate", "physics", "biology", "insect", "fish", "computer hardware", "plant", "sports team", "ecosystem", "reef", "wetland", "grassland", 'vehicle', 'airplane', 'bicycle', "animal", "chemical compound", "astronomical object", 'fruit', 'vegetable', 'cuisine', "planet", "physics", "chemistry", "mathematics", "biology", "geology", "ecology", "genetics", "space mission", "spacecraft", "particle", "species", "ecosystem", "hypothesis".
> - Although our datasets were not explicitly designed to account for the distribution of topic popularity, certain topics, such as diseases and volcanoes, contain more long-tail knowledge. However, our primary focus remains on evaluating knowledge editing methods, and our findings appear invariant to topics.
>
> \
> *Since the dataset is built from Wikipedia/Wikidata, which are frequently updated, there is potential for temporal mismatches between the dataset and the LLMs. For instance, some LLMs may have been trained on earlier versions, while others may align with more recent data. The authors might need to quantify and analyze the potential extent of outdated or inconsistent information across different LLMs to strengthen the paper’s validity, or propose a method to estimate the temporal alignment between the dataset and each LLM's training data.*
> - Investigating temporal mismatches is an interesting direction; however, evaluating temporal alignment is challenging because models often do not disclose the version of Wikidata used for training. Our main focus is on the improvements brought by knowledge editing.
> - Our results suggest that the findings are applicable across different LLMs, regardless of the version of Wikidata used for training.

---

> > ### Author Response · Authors · 2024-11-22
> >
> > *As mentioned in the paper, the hallucinations can happen differently for (1) different models, (2) pre-trained/fine-tuned in different periods, (3) different prompt input, etc. The generalization ability of the benchmark may be limited. Since LLMs' knowledge evolves with new releases, a focus on the dataset construction pipeline could add long-term value beyond the static dataset. The authors should emphasize the utility of their dataset construction pipeline, which might hold greater long-term value than the static dataset itself. I would suggest the authors to provide a detailed description of their dataset construction pipeline, including any code or tools used, to enable reproducibility and extension of their work*
> > - Due to space limitations, details such as topic selection were omitted in the initial draft to prioritize discussions on the five facets of editing methods. We have now included more details in Section 2 and our Anonymous GitHub (https://anonymous.4open.science/r/hallucination-9D6).
> > - Our data construction pipeline is designed to be easily adaptable to new models. Additional datasets, such as those for GPT-J and Gemma, have been added to our GitHub repository. We remain committed to releasing datasets for upcoming models to ensure reproducibility and extensibility.
> >
> > \
> > *In the dataset construction section, it is mentioned that Wikipedia was selected as the factual knowledge source, yet the triples were said to be retrieved using the Wikidata Query Service. Given that Wikipedia and Wikidata are distinct resources with content gaps, could the authors clarify where the dataset primarily comes from? Also, since Wikidata is actively edited, please report the version of the dump used for reproducibility.*
> > - All factual data were retrieved using the Wikidata Query Service with SPARQL. We have revised the manuscript to replace mentions of Wikipedia with Wikidata for consistency.
> > - The date of the Wikidata used has been reported in the updated manuscript in Section 2.1.
> >
> > \
> > *For completeness, when citing FT-L, it would be helpful to include both Meng et al., 2022, and the original work by Zhu et al., 2020.*
> > - The references for FT-L have been updated as suggested.
> >
> > \
> > *Given that 5 out of the 7 knowledge editing techniques evaluated have reported results on GPT family models in their original studies,  including at least one GPT model in this study would provide a more comprehensive baseline and strengthen the comparability of results.*
> > - We have added datasets for GPT-J to our GitHub repository to strengthen the baseline and ensure comparability.

---

> > > ### Comment · Reviewer_Umf8 · 2024-11-27
> > >
> > > Thank you for your response and edits. Once again, I find this research valuable, novel, and a meaningful contribution to the community. I highly recommend that the authors thoroughly document the pipeline to enable others to easily reproduce the dataset construction and results, which would enhance the impact of this work.
> > >
> > > Additionally, it appears that the results for GPT-J are not available either in the repository or in the paper itself.

---

> ### Author Response · Authors · 2024-12-03
> **Thanks for your comment! We have added the results of GPT-J and provided more explanations on our efforts for full reproducibility**
>
> Dear Rewer Umf8,
>
> We would like to express our sincere gratitude for your acknowledgement of our contributions. We are more than happy to provide more explanations as follows:
>
> > I highly recommend that the authors thoroughly document the pipeline to enable others to easily reproduce the dataset construction and results, which would enhance the impact of this work.
>
> We appreciate the comment and would like to emphasize that **we have tried our best to make results and dataset construction process fully reproducible**:
> - We have illustrated the details of three key steps of our dataset construction in Section 2.1 including: filtering hallucinations from LLMs based on Wikidata Query Service, removing triplets with the same subject and relation, and generating factual questions from triplets.
> - We have provided more details of the adopted LLMs in our paper and the process of generating evaluation questions in “Appendix A Reproducibility Statement”. More examples of the dataset are in “Appendix F Examples of HalluEditBench”.
> - Following your suggestion, we have provided more details on the selection process of topics and domains in the response above.
> - We have provided the code for dataset construction in https://anonymous.4open.science/r/hallucination-9D6/code/eval_hallu.py and the code for benchmarking all methods in https://anonymous.4open.science/r/hallucination-9D6/code/edit_all_method.py for full reproduction.
> - We have released all our experiment results in https://anonymous.4open.science/r/hallucination-9D6/results for verification of our findings.
>
> Any more specific suggestions to improve the reproducibility are greatly appreciated.
>
> > Additionally, it appears that the results for GPT-J are not available either in the repository or in the paper itself.
>
> Thanks for the suggestion. We have added the complete hallucination dataset of GPT-J in https://anonymous.4open.science/r/hallucination-9D6/data/questions/hallucination_final/gpt_j_6b
> (In addition, we have added the hallucination dataset of Gemma-2B in https://anonymous.4open.science/r/hallucination-9D6/data/questions/hallucination_final/gemma_1.1_2b_it for future studies)
>
> The complete results of GPT-J are released in https://anonymous.4open.science/r/hallucination-9D6/results/hallu_edit/gpt_j_6b, https://anonymous.4open.science/r/hallucination-9D6/results/hallu_edit_multi_hop/gpt_j_6b, and https://anonymous.4open.science/r/hallucination-9D6/results/hallu_edit_multi_turn/gpt_j_6b_multi_turn
>
> The dataset statistics and the figures for the results of GPT-J in five dimensions including Efficacy, Generalization, Portability, Locality, and Robustness are shown in https://anonymous.4open.science/r/hallucination-9D6/additional_experiments.md
>
> Based on the figures, **we can clearly see that the additional results of GPT-J further strengthen our three core findings**:
> - The effectiveness of knowledge editing methods in correcting real-world hallucinations could be far from what their performance on existing datasets suggests,
> - No editing methods can outperform others across five facets and the performance beyond Efficacy for all methods is generally unsatisfactory.
> - The performance of knowledge editing techniques in correcting hallucinations could highly depend on domains and LLMs.
>
> We are genuinely grateful for your valuable feedback and the acknowledgement of our contributions. We are willing to provide more explanations if you have any more questions.
>
> If you have already found our responses satisfactory, we humbly remind you of a fitting update of the final rating. Thanks for your time and effort again!
>
> The authors

---

> ### Author Response · Authors · 2024-12-03
> **Dear Reviewer, if you have already found our responses satisfactory, we humbly remind you of a fitting update of the final rating. Thanks for your time and effort again!**
>
> Dear Reviewer Umf8,
>
> We are genuinely grateful for your constructive feedback and the acknowledgement of our contributions. If you have already found our responses satisfactory, we humbly remind you of a fitting update of the final rating. Thanks for your time and effort again!
>
> The authors

---

### Meta-Review · Area_Chair_PziR · 2024-12-21

**Metareview:**

This paper presents HalluEditBench, a benchmarking framework that evaluates knowledge editing methods. Overall the reviews are positive because the paper makes fundamental contributions to the study of LLM hallucination correction and provides extensive experiments with insightful results showing how the effectiveness of knowledge editing techniques can vary significantly. There were also concerns on the description of methods and the depth of analysis, but these have been sufficiently addressed in the responses based on the reviewer discussion afterwards. I therefore recommend accepting the paper.

**Additional Comments On Reviewer Discussion:**

There are two major concerns on the paper, but they have been addressed as follows.
- Description of methods: reviewer Umf8 asked for a comprehensive description of the dataset construction process, and reviewer jvnV asked for a better introduction of the five scores. Both comments have been sufficiently addressed during the rebuttal.
- Depth of analysis: reviewer jvnV questioned the depth of analysis, and the authors gave a detailed response about their findings. Reviewer jvnV remains neutral, but did already increase his/her score once.

---

### Decision · Program_Chairs · 2025-01-22

Accept (Poster)